# CLIP-QDA: An Explainable Concept Bottleneck Model

**Rémi Kazmierczak**  *remi.kazmierczak@ensta-paris.fr*
*Unité d'Informatique et d'Ingénierie des Systèmes*
*ENSTA Paris, Institut Polytechnique de Paris*

**Eloïse Berthier**  *eloise.berthier@ensta-paris.fr*
*Unité d'Informatique et d'Ingénierie des Systèmes*
*ENSTA Paris, Institut Polytechnique de Paris*

**Goran Frehse**  *goran.frehse@ensta-paris.fr*
*Unité d'Informatique et d'Ingénierie des Systèmes*
*ENSTA Paris, Institut Polytechnique de Paris*

**Gianni Franchi**  *gianni.franchi@ensta-paris.fr*
*Unité d'Informatique et d'Ingénierie des Systèmes*
*ENSTA Paris, Institut Polytechnique de Paris*

**Reviewed on OpenReview:** *https://openreview.net/forum?id=jjmdiMiag7*

## Abstract

In this paper, we introduce an explainable algorithm designed from a multi-modal foundation model, that performs fast and explainable image classification. Drawing inspiration from CLIP-based Concept Bottleneck Models (CBMs), our method creates a latent space where each neuron is linked to a specific word. Observing that this latent space can be modeled with simple distributions, we use a Mixture of Gaussians (MoG) formalism to enhance the interpretability of this latent space. Then, we introduce CLIP-QDA, a classifier that only uses statistical values to infer labels from the concepts. In addition, this formalism allows for both sample-wise and dataset-wise explanations. These explanations come from the inner design of our architecture, our work is part of a new family of greybox models, combining performances of opaque foundation models and the interpretability of transparent models. Our empirical findings show that in instances where the MoG assumption holds, CLIP-QDA achieves similar accuracy with state-of-the-art CBMs. Our explanations compete with existing XAI methods while being faster to compute.

## 1 Introduction

The field of artificial intelligence is advancing rapidly, driven by sophisticated models like Deep Neural Networks (LeCun et al., 2015) (DNNs). These models find extensive applications in various real-world scenarios, including conversational chatbots (Ouyang et al., 2022), neural machine translation (Liu et al., 2020), and image generation (Rombach et al., 2021). Although these systems demonstrate remarkable accuracy, the process behind their decision-making often remains obscure. Consequently, deep learning encounters certain limitations and drawbacks. The most notable one is the lack of transparency regarding their behavior, which leaves users with limited insight into how specific decisions are reached. This lack of transparency becomes particularly problematic in high-stakes situations, such as medical diagnoses or autonomous vehicles.

The imperative to scrutinize the behavior of DNNs has become increasingly compelling as the field gravitates towards methods of larger scale in terms of both data utilization and number of parameters involved,

culminating in what is commonly referred to as "foundation models" (Brown et al., 2020; Radford et al., 2021; Ramesh et al., 2021). These models have demonstrated remarkable performance, particularly in the domain of generalization, while concurrently growing more intricate and opaque. Additionally, there is a burgeoning trend in the adoption of multimodality (Reed et al., 2022), wherein various modalities such as sound, image, and text are employed to depict a single concept. This strategic use of diverse data representations empowers neural networks to transcend their reliance on a solitary data format. Nonetheless, the underlying phenomena that govern the amalgamation of these disparate inputs into coherent representations remain shrouded in ambiguity and require further investigation.

The exploration of latent representations is crucial for understanding the internal dynamics of a DNN. DNNs possess the capability to transform input data into a space, called latent space, where inputs representing the same semantic concept are nearby. For example, in the latent space of a DNN trained to classify images, two different images of cats would be mapped to points that are close to each other (Johnson et al., 2016). This capability is further reinforced through the utilization of multimodality (Akkus et al., 2023), granting access to neurons that represent abstract concepts inherent to multiple types of data signals.

A class of networks that effectively exploits this notion is Concept Bottleneck Models (CBMs) (Koh et al., 2020). CBMs are characterized by their deliberate construction of representations for high-level human-understandable concepts, frequently denoted as words. Remarkably, there is a growing trend in employing CLIP (Radford et al., 2021), a foundation model that establishes a shared embedding space for both text and images, to generate concept bottleneck models in an unsupervised manner.

Unfortunately, while CLIP embeddings represent tangible concepts, the derived values, often termed "CLIP scores" pose challenges in terms of interpretation. Furthermore, to the best of our knowledge, there is a notable absence of studies that seek to formally characterize CLIP's latent space. The underlying objective here is to gain insights into how the pre-trained CLIP model organizes a given input distribution. Consequently, there is an opportunity to develop mathematically rigorous methodologies for elucidating the behavior of CLIP.

Then, our contributions are summarized as follows:

- We propose to represent the distribution of CLIP scores by a mixture of Gaussians. This representation enables a mathematically interpretable classification of images using human-understandable concepts.

- Utilizing the modeling approach presented in this study, we use Quadratic Discriminant Analysis (QDA) to classify the labels from the concepts, we name this method CLIP-QDA. CLIP-QDA demonstrates competitive performance when compared to existing CBMs based on CLIP. Notably, CLIP-QDA achieves this level of performance while utilizing a reduced set of parameters, limited solely to statistical values, including means, covariance matrices, and label probabilities.

- We propose two efficient and mathematically grounded XAI methods for model explanation, named CLIP-QDA$^{local}$ and CLIP-QDA$^{global}$. These methods encompass both global and local assessments of how the model behaves. The global metric directly emanates from our Gaussian modeling approach, providing a comprehensive evaluation of CLIP-QDA's performance. Additionally, our local metric draws inspiration from counterfactual analysis, furnishing insights into individual data points.

- We extend two established post-hoc XAI methods, LIME and SHAP, to formulate a novel XAI approach specifically tailored for CBMs. Departing from the conventional application of these methods, which typically produce explanations on the image level, CLIP-LIME and CLIP-SHAP generate explanations on the concept level allowed by CBMs.

- We propose a new evaluation protocol, specifically designed for the unique characteristics of CBMs to assess the effectiveness of explanations. This protocol includes a deletion metric, which examines faithfulness to the model, and a detection metric, which evaluates faithfulness to the data.

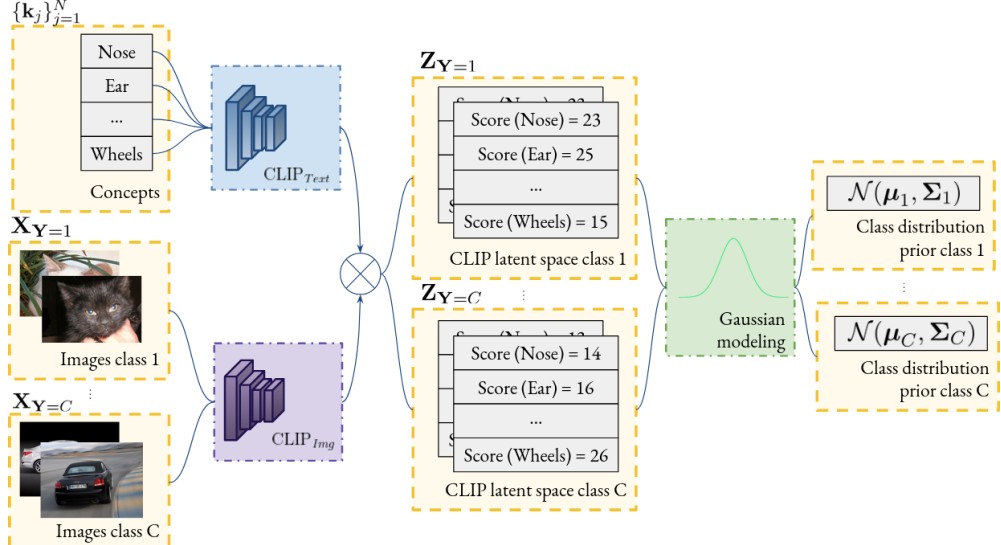

Figure 1: **Overview of our modeling method.** By considering the whole dataset CLIP scores $\boldsymbol{z}$ as class conditioned distributions $\boldsymbol{Z} = \begin{bmatrix} Z^1 & \dots & Z^N \end{bmatrix}$, we model the CLIP latent space as a mixture of Gaussians, allowing for mathematically grounded explanations.

## 2 Background and Related Work

### 2.1 Contrastive Image Language Pre-training (CLIP)

CLIP (Radford et al., 2021) is a multi-modal model that can jointly process image and text inputs. The model was pre-trained on a large dataset of images and texts to learn to associate visual and textual concepts. Then, the capacity of CLIP to create a semantically rich encoding induced the creation of many emergent models in detection, few-shot learning, or image captioning.

The widespread adoption of CLIP stems from the remarkable robustness exhibited by its pre-trained model. Through training on an extensive multimodal dataset, such as Schuhmann et al. (2022), the model achieves impressive performance. Thus, on few- and zero-shot learning, for which it was designed, it obtains impressive results across a wide range of datasets. Notably, CLIP provides a straightforward and efficient means of obtaining semantically rich representations of images in low-dimensional spaces. This capability enables researchers and practitioners to divert the original use of CLIP to various other applications (Luo et al., 2022; Menon & Vondrick, 2022; Gabeff et al., 2023).

### 2.2 CLIP-based Concept Bottleneck Models (CLIP-based CBMs)

The term Concept Bottleneck Model (CBM), as outlined in Koh et al. (2020), refers to to the implementation of a bottleneck reliant on human-specified concepts to execute a task, predominantly image classification. Consequently, the resultant algorithm inherently facilitates enhanced interpretability. While the term itself is relatively recent, it characterizes a lineage of methods that were employed in earlier research (Kumar et al., 2009; Lampert et al., 2009; Koh et al., 2020; Losch et al., 2019). However, despite the advantages offered by CBMs in terms of better understanding, early implementations encountered challenges stemming from the requirement for dedicated datasets. These datasets needed to encompass not only inputs and labels but also incorporate human-specified concepts for each sample.

In this context, the emergence of multimodal foundational models has opened up novel opportunities. Recent research (Yang et al., 2023; Oikarinen et al., 2023) has leveraged large language models to directly construct concepts from CLIP text embeddings, opening the door to a family of CLIP-based CBMs. Additionally, efforts have been made to create sparse CLIP-based CBMs (Panousis et al., 2023; Feng et al., 2023). Yan

et al. (2023a) explore methods to achieve superior representations with minimal labels. Yuksekgonul et al. (2022) capitalize on the CLIP embedding spaces, considering concepts as activation vectors. Finally, Kim et al. (2023) build upon the idea of activation vectors to discover counterfactuals.

### 2.3 Explainable AI

According to Arrieta et al. (2020), we can define an explainable model as a computational model, that is designed to provide specific details or reasons to ensure clarity and ease of understanding regarding its functioning. In broader terms, an explanation denotes the information or output that an explainable model delivers to elucidate its operational processes.

The literature shows a clear distinction between non-transparent (or blackbox) and transparent (or whitebox) models. Transparent models are characterized by their inherent explainability. These models can be readily explained due to their simplicity and easily interpretable features. Examples of such models include linear regression (Galton, 1886), logistic regression (McCullagh, 2019), and decision trees (Quinlan, 1986). In contrast, non-transparent models are inherently non-explainable. This category encompasses models that could have been explainable if they possessed simpler and more interpretable features (Galton, 1886; Quinlan, 1986), as well as models that inherently lack explainability, including deep neural networks. The distinction between these two types of models highlights the trade-off between model complexity and interpretability (Arrieta et al., 2020), with transparent models offering inherent explainability while non-transparent models allow for better performance but require the use of additional techniques for explanations, named post-hoc methods. Commonly used post-hoc methods include visualization techniques, such as saliency maps, which highlight the influential features in an image that contribute to the model's decision-making. Within this category of methods, notable examples include approaches such as Grad-CAM (Selvaraju et al., 2017), which generates activation maps by computing the gradients of the output labels. Sensitivity analysis (Cortez & Embrechts, 2011) represents another avenue, involving the analysis of model predictions by varying input data. Sample-wise explanation techniques are also used to explain the model from a local simplification of the model around a point of interest (Ribeiro et al., 2016; Plumb et al., 2018). Finally, feature relevance techniques aim at estimating the impact of each feature on the decision (Lundberg & Lee, 2017).

In an endeavor to integrate the strengths of both black and whitebox models, the concept of greybox XAI has been introduced by Bennetot et al. (2022). These models divide the overall architecture into two distinct components. Initially, a blackbox model is employed to process high-entropy input signals, such as images, and transform them into a lower-entropy latent space that is semantically meaningful and understandable by humans. By leveraging the blackbox model's ability to simplify complex problems, a whitebox model is then used to deduce the final result based on the output of the blackbox model. This approach yields a partially explainable model that outperforms traditional whitebox models while retaining partial transparency, in a unified framework.

Feature Attribution Methods, a category of techniques employed to address the complexity of DNNs' output for explainability, strive to identify crucial features in the input. These methods leverage a mapping function to reduce input complexity. Notable examples in this family include DeepSHAP and KernelSHAP, as proposed by Lundberg & Lee (2017). This approach resonates with the concept of greybox models, wherein the input is initially simplified to be explainable by a transparent classifier. However, greyboxes differ from Feature Attribution Methods in that the mapping function is independent from the input under consideration.

## 3 A Greybox Concept Bottleneck Model: CLIP-QDA

### 3.1 General Framework

For our experimental investigations, we consider a general framework based on prior work on CLIP-based CBMs (Yang et al., 2023; Oikarinen et al., 2023). This framework consists of two core components. The first component centers on the extraction of multi-modal features, enabling the creation of connections between

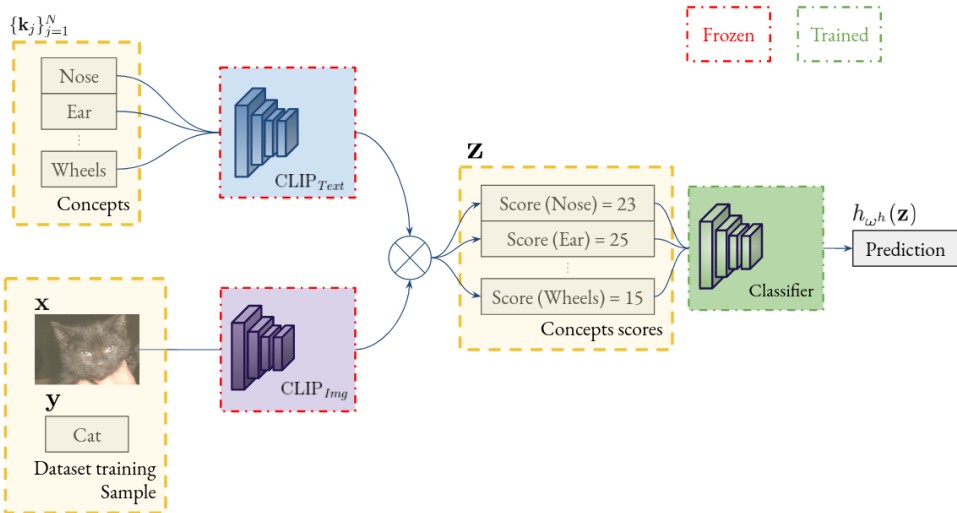

Figure 2: **Training procedure of the general framework.** First, CLIP scores $\boldsymbol{z}$ are computed for each of the concepts $\{\boldsymbol{k}^j\}_{j=1}^N$, then a classifier $h_{\boldsymbol{\omega}^h}(.)$, with parameters $\boldsymbol{\omega}^h$ is trained to classify the label from the concatenation of the CLIP scores.

images and text. The second component encompasses a classifier head. A visual depiction of this process is presented in Figure 2.

We build upon CLIP DNN (Radford et al., 2021), which enables the creation of a multi-modal latent space through the fusion of image and text information. Rather than relying on a single text or prompt, we employ a set of diverse prompts, each representing distinct concepts. These concepts remain consistent across the dataset and are not subject to alterations. The purpose of CLIP's representation is to gauge the similarities between each concept and an image, thereby giving rise to a latent space. To prevent ambiguity, we denote the resulting space of CLIP scores as the "CLIP latent space", while the spaces generated by the text and image encoders are respectively referred to as the "CLIP text embedding space" and the "CLIP image embedding space". Here, "CLIP score" denotes the value derived from a cosine distance computation between the image and text encodings.

The selection of concepts is guided by expert input and acts as a hyperparameter within our framework. For comprehensive examples of concept sets, please refer to Section A.3. Notably, there is no requirement for individual image annotation with these concepts. This is due to CLIP's inherent design, which allows it to score concepts in a zero-shot manner.

Following the acquisition of the CLIP latent space, it is given as an input to a classifier head, which is responsible for learning to predict the class. Thanks to the low dimension of the latent space and the clear semantics of each component (concepts), it is possible to design simple and explainable classifiers.

### 3.2 CLIP Latent Space Analysis

#### 3.2.1 Notations and formalism

Let us introduce the following notations used in the rest of the paper. $X$ and $Y$ represent two random variables (RVs) with joint distribution $(X, Y) \sim \mathcal{P}_{X,Y}$. A realization of this distribution is a pair $(\boldsymbol{x}, y)$ that concretely represents one image and its label. In particular, $y$ takes values in $[\![1, C]\!]$, with $C \in \mathbb{N}$ the number of classes. From this distribution, we can deduce the marginal distributions $X \sim \mathcal{P}_X$ and $Y \sim \mathcal{P}_Y$. We can also describe for each class $c$, an RV $X_{Y=c} \sim \mathcal{P}_{X_{Y=c}}$ that represents the conditional distribution of images that have the class $c$.

Let $\{\boldsymbol{k}^j\}_{j=1}^N$ denote a set of $N \in \mathbb{N}$ concepts, where each $\boldsymbol{k}^j$ is a character string representing the concept in natural language. We consider "CLIP's DNN" to refer to the vector of its pre-trained weights, denoted as $\boldsymbol{\omega}^g$, and a function $g$ that represents the architecture of the deep neural network (DNN). Given an image $\boldsymbol{x}$ and a concept $\boldsymbol{k}^j$, the output of CLIP's DNN is represented as $z^j = g_{\boldsymbol{\omega}^g}(\boldsymbol{x}, \boldsymbol{k}^j)$. The projection in the multi-modal latent space of an image $\boldsymbol{x}$ is the vector $\boldsymbol{z} = \begin{bmatrix} g_{\boldsymbol{\omega}^g}(\boldsymbol{x}, \boldsymbol{k}^1) & \dots & g_{\boldsymbol{\omega}^g}(\boldsymbol{x}, \boldsymbol{k}^N) \end{bmatrix}$. We define $Z^j$ as the random variable associated with the observation $z^j$. It should be noted that $\boldsymbol{Z} = \begin{bmatrix} Z^1 & \dots & Z^N \end{bmatrix}$ is the random variable representing the concatenation of the CLIP scores associated with the $N$ concepts. Furthermore, we denote the conditional distributions of $\boldsymbol{z}$ having class $c$ as $\boldsymbol{Z}_{Y=c} = \begin{bmatrix} Z_{Y=c}^1 & \dots & Z_{Y=c}^N \end{bmatrix}$.

Finally, we define the classifier as a function $h_{\boldsymbol{\omega}^h}(\boldsymbol{z})$ with parameters $\boldsymbol{\omega}^h$ that, given a vector $\boldsymbol{z}$, outputs the predicted class.

### 3.2.2 Gaussian modeling of CLIP's latent space

To analyze the behavior of the CLIP latent space, we conduct a thorough examination of the distribution of CLIP scores. To elucidate our modeling approach, we suggest to visualize a large set of samples from $Z^j$ by observing the CLIP scores of an entire set of images taken from a toy example, which consists of images representing only cats and cars (see the Cats/Dogs/Cars dataset in Section 4). In this instance, the concept denoted by "$j$" corresponds to "Pointy-eared".

In Figure 3, which illustrates the histogram of CLIP scores, we observe that the distribution exhibits characteristics that can be effectively modeled as a mixture of two Gaussians. The underlying intuition here is that the distribution $Z^j$ represents two types of images: those without pointy ears, resulting in the left mode (low scores) of CLIP scores, and those with pointy ears, resulting in the right mode (high scores). Since this concept uniquely characterizes the classes – cats have ears but not cars – we can assign each mode to a specific class. This intuition is corroborated by the visualization of the distribution $Z_{Y=1}^j$ (Car) in violet and the distribution $Z_{Y=2}^j$ (Cat) in red. Since the extracted distributions exhibit similarities to normal distributions, we are motivated to describe $\boldsymbol{Z}$ as a mixture of Gaussians. Yet, we also discuss the validity and limitations of this modeling approach in Section 5.1.

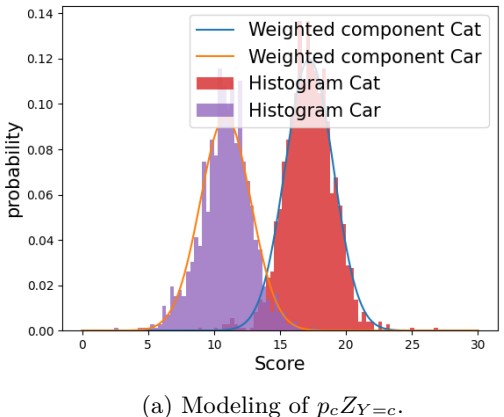
(a) Modeling of $p_c Z_{Y=c}$.

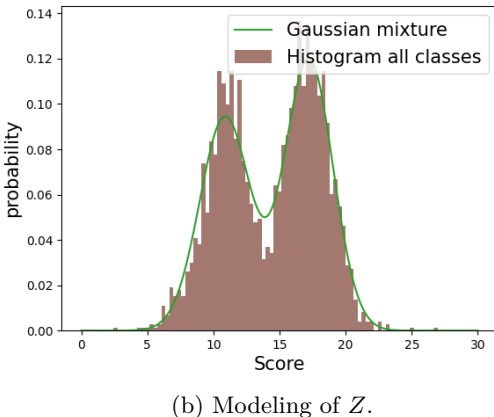
(b) Modeling of $Z$.

Figure 3: **Normalized histogram of scores $Z^j$ specifically for the concept "Pointy-eared".** On the left, we observe that the different classes can be modeled as weighted Gaussians. On the right, we show the resulting Gaussian mixture modeling.

Mathematically, the Gaussian prior assumption is equivalent to:

$$p(\boldsymbol{Z} = \boldsymbol{z} \mid Y = c) = \mathcal{N}(\boldsymbol{z} \mid \boldsymbol{\mu}_c, \boldsymbol{\Sigma}_c), \tag{1}$$

where $\boldsymbol{\Sigma}_c$ and $\boldsymbol{\mu}_c$ are the mean vectors and the covariance matrices, different for each class. Moreover, given the multinomial distribution of $Y$, with the notation $p_c = P(Y = c)$, we can model $\boldsymbol{Z}$ as a mixture of

Gaussians:

$$p(\boldsymbol{Z} = \boldsymbol{z}) = \sum_{c_i=1}^{C} p_{c_i} \mathcal{N}(\boldsymbol{z} \mid \boldsymbol{\mu}_{c_i}, \boldsymbol{\Sigma}_{c_i}) \ . \tag{2}$$

### 3.3 CLIP Quadratic Discriminant Analysis (CLIP-QDA)

Based on the Gaussian distribution assumption described in Section 3.2.2, a natural choice for $h_{\boldsymbol{\omega}^h}$ (the classifier in Figure 2) is the Quadratic Discriminant Analysis (QDA) as defined in Hastie et al. (2009). To compute it, we need to estimate the parameters $(\boldsymbol{\Sigma}_c, \boldsymbol{\mu}_c, p_c)$ of the probability distributions $\boldsymbol{Z}_{Y=c}$ and $Y$, which is done by computing the maximum likelihood estimators on the training data.

Subsequently, with the knowledge of the functions $p(\boldsymbol{Z} = \boldsymbol{z} \mid Y = c)$ and $p(Y = c)$, we can apply Bayes theorem to make an inference on $p(Y = c \mid \boldsymbol{Z} = \boldsymbol{z})$:

$$p(Y = c \mid \boldsymbol{Z} = \boldsymbol{z}) = \frac{p_c \mathcal{N}(\boldsymbol{z} \mid \boldsymbol{\mu}_c, \boldsymbol{\Sigma}_c)}{\sum_{c_i=1}^{N} p_{c_i} \mathcal{N}(\boldsymbol{z} \mid \boldsymbol{\mu}_{c_i}, \boldsymbol{\Sigma}_{c_i})} \ . \tag{3}$$

Then, the output of the QDA classifier can be described as:

$$h_{\boldsymbol{\omega}^h}(\boldsymbol{z}) = \arg\max_{c} \ \ \frac{p_c}{(2\pi)^{N/2}|\boldsymbol{\Sigma}_c|^{1/2}} e^{-\frac{1}{2}(\boldsymbol{z}-\boldsymbol{\mu}_c)^T \boldsymbol{\Sigma}_c^{-1}(\boldsymbol{z}-\boldsymbol{\mu}_c)} \ . \tag{4}$$

In practice, we leverage the training data to estimate $\boldsymbol{\omega}^h = (\boldsymbol{\Sigma}_c, \boldsymbol{\mu}_c, p_c)$, which enables us to bypass the standard stochastic gradient descent process, resulting in an immediate "training time". Furthermore, this classifier offers the advantage of transparency, akin to the approach outlined by Arrieta et al. (2020), with its parameters comprising identifiable statistical values and its output values representing probabilities.

### 3.4 Explainable AI for Concept Bottleneck Models

Our CLIP-QDA model is founded upon a transparent probabilistic framework, hence, we have at our disposal a variety of statistical tools to explain the functioning of our classifier as illustrated in Figure 4. In this section, we present two distinct types of explanations: CLIP-QDA$^{global}$, offering a *global* perspective that sheds light on the classifier's behavior across the entire dataset (refer to Section 3.4.1), and CLIP-QDA$^{local}$, providing a *local* explanation tailored to elucidate the model's actions on individual samples (refer to Section 3.4.2).

Furthermore, we propose the adaptation of two well-established XAI post-hoc methods, LIME (Ribeiro et al., 2016) and SHAP (Lundberg & Lee, 2017), to the unique characteristics of CBMs. These methods are denoted as CLIP-SHAP and CLIP-LIME, respectively (see Section 3.4.3). An overview of the methods incorporated in our investigation is depicted in Figure 4.

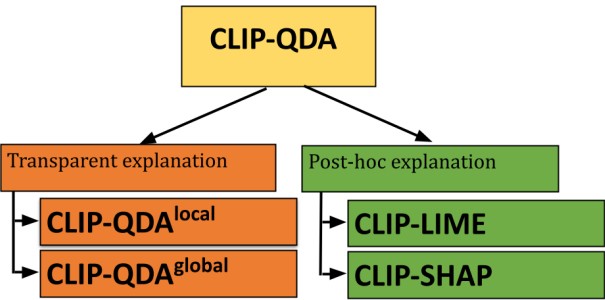

Figure 4: **The different source of explanation of CLIP-QDA.** Global (CLIP-QDA$^{global}$) vs. Local (CLIP-QDA$^{local}$) explanations offer insights into classifier behavior across the dataset and on individual samples, respectively. Post-hoc explanations with CLIP-SHAP and CLIP-LIME use traditional XAI techniques.

### 3.4.1 Dataset-wise explanation with CLIP-QDA$^{global}$

As we have access to priors that describe the distribution of each class, a valuable insight to gain an understanding of which concept our classifier aligns with is the measurement of distances between these distributions. Specifically, we focus on the conditional distributions of two classes of interest $c_1$ and $c_2$, that we denote by $Z^j_{Y=c_1}$ and $Z^j_{Y=c_2}$. The underlying intuition behind measuring the distance between these distributions is that the larger the distance, the more the attribute $j$ can differentiate between the classes $c_1$ and $c_2$.

To accomplish this, we propose to use the Wasserstein-2 distance (Ramdas et al., 2017) as a metric for quantifying the separation between the two conditional distributions. It is worth noting that calculating the Wasserstein-2 distance can be a complex task in general. However, for Gaussian distributions, there exists a closed-form solution for computing the Wasserstein-2 distance. In addition, we sign the distance to keep the information of the position of $c_1$ relative to $c_2$:

$$\tilde{W}_2(Z^j_{Y=c_1}, Z^j_{Y=c_2}) = \text{sign}([\boldsymbol{\mu}_{c_1}]_{(j)} - [\boldsymbol{\mu}_{c_2}]_{(j)}) \left( ([\boldsymbol{\mu}_{c_1}]_{(j)} - [\boldsymbol{\mu}_{c_2}]_{(j)})^2 + \Lambda^j_{c_1,c_2} \right),$$

where $\Lambda^j_{c_1,c_2} = [\boldsymbol{\Sigma}_{c_1}]_{(j,j)} + [\boldsymbol{\Sigma}_{c_2}]_{(j,j)} - 2\sqrt{[\boldsymbol{\Sigma}_{c_1}]_{(j,j)}[\boldsymbol{\Sigma}_{c_2}]_{(j,j)}}$.

Note that the resulting value is no longer a distance since we lost the commutativity property. Examples of explanations based on this metric are given in Sections 5.4 and A.7. Also, an alternative way to produce dataset-wise explanations, oriented on example-based explanations, is also available in Section A.4.

### 3.4.2 Sample-wise explanation with CLIP-QDA$^{local}$

One would like to identify the key concepts associated with a particular image that plays a pivotal role in achieving the task's objective. To delve deeper into the importance and relevance of concepts in the decision-making process of the classifier, a widely accepted approach is to generate counterfactuals (Plumb et al., 2022; Luo et al., 2023; Kim et al., 2023). If a small perturbation of a concept score changes the class, the concept is considered important. We now formalize this mathematically.

Consider a pre-trained classifier denoted as $h_{\boldsymbol{\omega}^h}(\cdot)$. In this context, $\boldsymbol{\omega}^h$ represents the set of weights associated with the CLIP-QDA, specifically $\boldsymbol{\omega}^h = (\boldsymbol{\Sigma}_c, \boldsymbol{\mu}_c, p_c)$. Given a score vector $\boldsymbol{z}$, we define counterfactuals as hypothetical values $\boldsymbol{z} + \boldsymbol{\epsilon}^j_s$, $\boldsymbol{\epsilon}^j_s$ being called the perturbation. This perturbation aims to be of minimal magnitude and is obtained by solving the following optimization problem:

$$\min \|\boldsymbol{\epsilon}^j_s\|^2 \quad \text{s.t.} \quad h_{\boldsymbol{\omega}^h}(\boldsymbol{z} + \boldsymbol{\epsilon}^j_s) \neq h_{\boldsymbol{\omega}^h}(\boldsymbol{z}). \tag{5}$$

The idea behind this equation is to find the minimal perturbation $\boldsymbol{\epsilon}^j_s$ of the input $\boldsymbol{z}$ that makes the classifier produce a different label than $h_{\boldsymbol{\omega}^h}(\boldsymbol{z})$. However, in our case, two important restrictions are applied to $\boldsymbol{\epsilon}^j_s$:

1. **Sparsity**: for interpretability, we only change one attribute at a time, indicated by the index $j$. Then $\boldsymbol{\epsilon}^j_s = [0, .., 0, \epsilon^j_s, 0, ..., 0]$.

2. **Sign**: we take into account the sign $s \in \{-, +\}$ of the perturbation. Then, we separate the positive counterfactuals $\boldsymbol{\epsilon}^j_+ = [0, .., 0, \epsilon^j_+, 0, ..., 0]$, $\epsilon^j_+ \in \mathbb{R}^+$ and the negative counterfactuals. $\boldsymbol{\epsilon}^j_- = [0, .., 0, \epsilon^j_-, 0, ..., 0]$, $\epsilon^j_- \in \mathbb{R}^-$.

These two constraints are imposed to generate concise and, consequently, more informative counterfactuals. In this context, if a solution to equation 5, denoted as $\boldsymbol{\epsilon}^j_{s,*}$, exists, it represents the minimal modification (addition or subtraction) to the coordinate $j$ of the original vector $\boldsymbol{z}$ that results in a change from $h_{\boldsymbol{\omega}^h}(\boldsymbol{z})$ to $h_{\boldsymbol{\omega}^h}(\boldsymbol{z} + \boldsymbol{\epsilon}^j_{s,*})$. Note that this approach allows for an explicit evaluation of the effect of an *intervention*, denoted as $do(\boldsymbol{Z} = \boldsymbol{z} + \boldsymbol{\epsilon}^j_s)$ using a common notation in causal inference (Peters et al., 2017). Concretely, this emulates answers to questions of the form: "Would the label of my cat's image change if I removed a certain amount of its pointy ears?".

Another important point to notice is that to obtain all possible counterfactuals, this equation must be solved for all concepts $j$ and both signs $s$. A practical way to compute counterfactuals is given below.

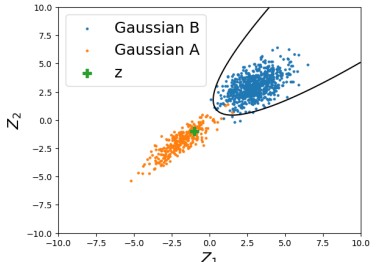
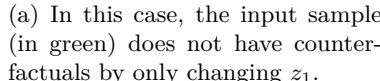
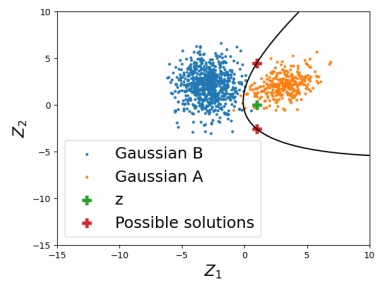
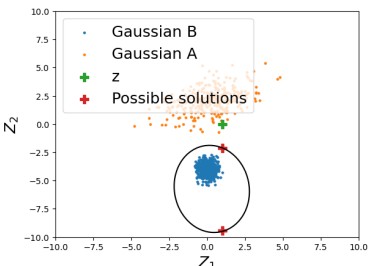

(a) In this case, the input sample (in green) does not have counterfactuals by only changing $z_1$.

(b) In this case, the input sample (in green) has counterfactuals $\boldsymbol{\epsilon}^1_{-,*}$ and $\boldsymbol{\epsilon}^1_{+,*}$. Changing $z_1$ leads to two intersections with the equiprobability line, one by adding score, the other by removing score.

(c) In this case, the input sample (in green) only has one counterfactual $\boldsymbol{\epsilon}^1_{-,*}$ and no counterfactual $\boldsymbol{\epsilon}^1_{+,*}$. Changing $z_1$ leads to two intersections with the equiprobability line, but since both are by removing score, only the one of minimal magnitude is conserved.

Figure 5: **Visualization of the counterfactuals in the two Gaussians toy example.** Samples of the two distributions are plotted in blue and orange. The equiprobability line is plotted in black.

**Proposition 1.** *Let us consider a pre-trained QDA classifier $h_{\boldsymbol{\omega}^h}(.)$ with parameters $\boldsymbol{\omega}^h$. Assume that the input data is drawn from the corresponding Gaussian Mixture model, as defined in 4, and that $\boldsymbol{\epsilon}^j_s$ a perturbation with the above sparsity and sign restrictions. Then, there is a closed-form solution to problem 5, which is a function of the parameters $(\boldsymbol{\Sigma}_c, \boldsymbol{\mu}_c, p_c)^C_{c=1}$.*

The proof and expression are given in Appendix A.5. We illustrate the behavior of our classifier and our *local* metric with a toy example which consists of two Gaussians ($C = 2$) among two concepts $Z_1$ and $Z_2$ ($N = 2$). We find the counterfactuals for both signs following the first concept ($\boldsymbol{\epsilon}^1_{-,*}$ and $\boldsymbol{\epsilon}^1_{+,*}$). Results are presented in Figure 5.

It is worth noting that these counterfactual values are initially expressed in CLIP score units, which may not inherently provide meaningful interpretability. To mitigate this limitation, we introduce scaled counterfactuals, denoted as $\boldsymbol{\epsilon}^j_{s,*,scaled}$, obtained by dividing each counterfactual by the standard deviation associated with its respective distribution:

$$\boldsymbol{\epsilon}^j_{s,*,scaled} = \frac{\boldsymbol{\epsilon}^j_{s,*}}{\sqrt{[\boldsymbol{\Sigma}_c]_{(j,j)}}} \, . \tag{6}$$

Then, the value of each counterfactual can be expressed as "the addition (or subtraction) of standard deviations in accordance to $\boldsymbol{Z}_{Y=h_{\boldsymbol{\omega}^h}(\boldsymbol{z})}$ that changes the label". Examples of such explanations are given in Sections 5.4 and A.7.

### 3.4.3 CLIP-LIME and CLIP-SHAP

**CLIP-LIME.** To adapt LIME to the operation of CBMs, we begin with the image input $x$. From this input, we calculate the projection in the latent space $\boldsymbol{z} = \begin{bmatrix} g_{\boldsymbol{\omega}^g}(\boldsymbol{x}, \boldsymbol{k}^1) & \dots & g_{\boldsymbol{\omega}^g}(\boldsymbol{x}, \boldsymbol{k}^N) \end{bmatrix}$. Subsequently, following the LIME method, we train a surrogate model to approximate $h_{\boldsymbol{\omega}^h}$ in the vicinity of $\boldsymbol{z}$ by training it on a dataset comprised of perturbed inputs around $\boldsymbol{z}$. Finally, the explanation is derived from the importance weights of the resulting surrogate model.

**CLIP-SHAP.** To adapt SHAP to CBMs, we also consider the projection $\boldsymbol{z} = \begin{bmatrix} g_{\boldsymbol{\omega}^g}(\boldsymbol{x}, \boldsymbol{k}^1) & \dots & g_{\boldsymbol{\omega}^g}(\boldsymbol{x}, \boldsymbol{k}^N) \end{bmatrix}$. To compute statistical values relevant to CLIP-SHAP, we incorporate the projections of all the images in the training set. Given that the number of concepts can be high, we employ the Kernel version of SHAP (Kernel SHAP) to reduce the computational cost. The resulting explanation comprises the Shapley values associated with each of the concepts.

## 4  Experimental Setup

### 4.1  Datasets

We evaluate our methods on ImageNet (Deng et al., 2009), PASCAL-Part (Donadello & Serafini, 2016), MIT Indoor Scenes dataset (Quattoni & Torralba, 2009), MonuMAI (Lamas et al., 2021) and Flowers102 (Nilsback & Zisserman, 2008). In addition to these well-established datasets, we introduce a custom dataset named Cats/Dogs/Cars dataset (Section A.2). To construct this dataset, we concatenated two widely recognized datasets, namely, the Kaggle Cats and Dogs Dataset (Cukierski, 2013) and the Standford Cars Dataset (Krause et al., 2013). Subsequently, we filtered the dataset to contain images of white and black animals and cars exclusively. This curation resulted in six distinct subsets: "Black Cars", "Black Dogs", "Black Cats", "White Cars", "White Dogs", "White Cats". The primary objective of this dataset is to facilitate experiments under conditions of substantial data bias, such as classifying white cats when the training data has only encountered white dogs and black cats. Specifically, we refer to two distinct scenarios (Table 1): one containing cats and cars of both colors (referred to as the unbiased setup) and the other one with only black cats and white cars (referred to as the biased setup). When none of theses scenarios are referred to (for example in the *Del* values of Table 4 and 5), the complete dataset is used. In its final form, the dataset comprises 6,436 images. Additional information is available in Section A.2.

Table 1: **Cats/Dogs/Cars compositions used in our study**

|  | Complete dataset | Biased setup | Unbiased setup |
|---|---|---|---|
| Composition | Black Cats, White Cats, Black Cars, White Cats, Black Dogs, White Dogs | Black Cats, White Cars | Black Cats, White Cats, Black Cars, White Cars |
| Num samples | 6436 | 2536 | 4031 |

### 4.2  Baselines

**Classifiers.** Here, we present the baseline used for comparing performance in terms of inference (i.e., accuracy) with other algorithms performing classification tasks. First, we evaluate various classifiers, which are represented as the trainable component illustrated in Figure 2. We denote the method proposed by Yan et al., which involves training a linear layer as a classifier from CLIP scores, as one of the baselines. As used notably in (Yan et al., 2023a;b), this approach is referred to as linear probe. Additionally, we adopt LaBo, introduced by Yang et al. (2023), which employs a class-concept matrix as the classifier of the CBMs. For all these methods, the same set of concepts is used, with a comprehensive description of the procedure for acquiring the concept set provided in Appendix A.3. We also incorporate previous CBMs such as Greybox XAI (Bennetot et al., 2022), X-NeSyL (Díaz-Rodríguez et al., 2022), and the method proposed by Morales Rodríguez et al. (2024) into our study, which provides explainability but requires additional training data related with annotated concepts. It is noteworthy that these methods, requiring training from images, entail significantly longer training times. Additionally, we assess methods widely used as classifiers that are not CBMs, including the use of CLIP as a zero-shot classifier. We also include ResNet (He et al., 2016) and ViT (Dosovitskiy et al., 2021) trained in a supervised manner using images as inputs. Further implementation details can be found in Section A.1.

**XAI Methods.** To compare our proposed XAI methods (CLIP-QDA$^{global}$, CLIP-QDA$^{local}$, CLIP-LIME, and CLIP-SHAP) with existing works, we evaluated several methods, including those providing explanations at both the image and concept levels. At the concept level, we examined explanations provided by LaBo (Yang et al., 2023) and Yan et al. (Yan et al., 2023b). For image-based explanations, we assessed SHAP (Lundberg & Lee, 2017) and LIME (Ribeiro et al., 2016). Further implementation details can be found in Section A.1.

### 4.3 CBM Quantitative Evaluation Process

Assessing the quality of explanations has long been a challenge, given its subjective nature. While quantitative evaluation processes exist for evaluating performance in a general setup (Hedström et al., 2023), they may not be well-suited to the specific characteristics of CBMs. Consequently, we present a novel method designed to evaluate XAI solutions, tailored to CBMs. This method comprises two metrics: the Deletion metric, gauging faithfulness to the model, and the Detection metric, quantifying faithfulness to the data.

**Deletion metric.** The deletion metric is adapted from the methodology introduced by Petsiuk et al. (2018). The procedure involves taking each sample from the test set and nullifying (i.e., setting to the average value of the score across classes) a certain number, $N_{null}$, of concepts. We nullify the concepts based on their importance order as determined by each explanation method. If nullifying the concepts leads to a significant decrease in performance, we consider it a successful selection of concepts that influenced the classifier's decision. The intuition behind this idea is that if the concept is important, its absence will result in a loss of performance. Given the different values of accuracy obtained by nullifying $N_{null}$ concepts, denoted by $Acc(N_{null})$, we deduce the deletion score $Del$ by computing the area under the curve of $Acc(N_{null})$. The interest is to probe the ability to correctly order the important concepts in the explanation:

$$Del = \frac{1}{Acc(0)} \sum_{i=1}^{N_{max}} \frac{Acc(i-1) + Acc(i)}{2} .$$  (7)

Here, $N_{max}$ represents a hyperparameter that defines the maximum number of deletions considered. The selection of this hyperparameter is critical: it balances between capturing the metric's capacity to fit the model and ensuring that the computed inputs remain plausible. A default value of 9 is assigned to maintain this equilibrium. Note that we normalize the result by the maximum accuracy $Acc(0)$ to allow for better comparisons among different classifiers.

The evaluation framework is structured into two setups: the first one uses the concept set outlined in Table 6 (referred to as Set 1), while the second one uses an equivalent number of concepts randomly chosen from a dictionary of words (referred to as Set 2).

**Detection metric.** The detection metric evaluates the model's ability to identify relevant concepts. For each sample $s$, a set of ground truth concepts $\mathcal{S}_s$ to detect is defined by an oracle. Then, we construct a set of concepts $\mathcal{T}_s$ consisting of the top $|\mathcal{S}_s|$ concepts in the explanation to be tested. The resulting detection metric is the average ratio of agreements among all samples in the test dataset:

$$Det = \frac{1}{S} \sum_{s=1}^{S} \frac{|\mathcal{S}_s \cap \mathcal{T}_s|}{|\mathcal{S}_s|},$$  (8)

where $S$ is the total number of samples in the dataset. The selection of ground truth concepts depends on the available options within each dataset, as elaborated in Section 5.3.

## 5 Experiments

### 5.1 Assessing the Gaussian Prior Hypothesis

In this section, we investigate to which extent the Gaussian prior hypothesis (Equation 1) holds. To assess this, we use Chi-Square Q-Q plots (Chambers, 2018; Mahalanobis, 2018), a normality assessment method adapted to multidimensional data. We display Chi-Square Q-Q plots on the conditional distribution $(\boldsymbol{Z} \mid Y = c)$, with data sourced from PASCAL-Part and class $c =$ "aeroplane". First, we compute a set of concepts adapted to the output class, following the procedure described in Appendix A.3. Subsequently, we perform the same experiment with a set of words specifically dedicated to the class of interest (Figure 6a). In addition, we show a visualization with randomly chosen concepts from the PASCAL-Part set of concepts, as depicted in Figure 6b.

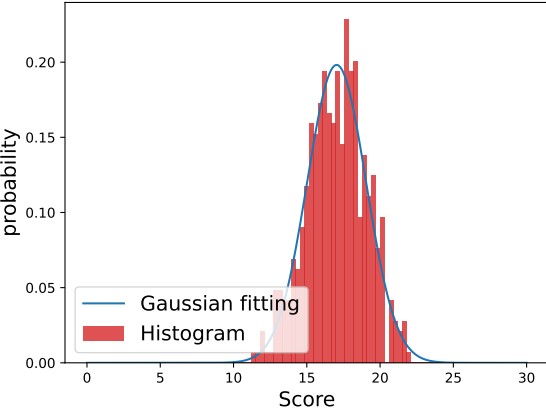

Figure 7: **Histogram and Gaussian fitting of CLIP scores $z$ of the attribute "Multi-doored", for images that have the class "aeroplane".**

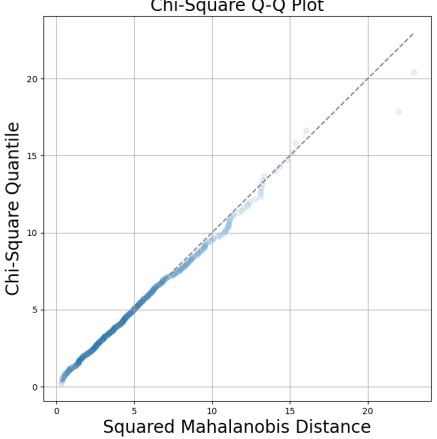

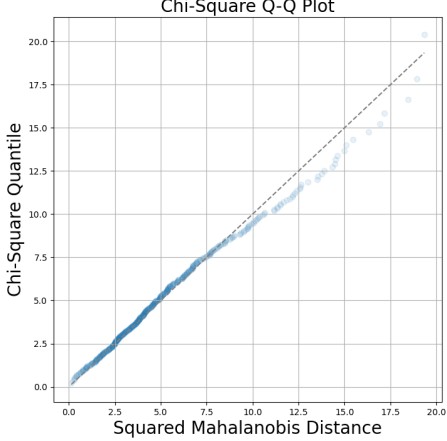

(a) **CBM with a set of concepts related to the label "Aeroplane"**: [Winged, Jet engines, Tail fin, Fuselage, Landing gear]

(b) **CBM with a set of concepts unrelated to the label "Aeroplane"**: [Furry, Equine, Container or pot, Saddle or seat, Multi-doored]

Figure 6: **Multivariate Q-Q plot illustrating the Gaussian fit for conditional modeling of the latent space, given a fixed label.** The plot is specifically for PASCAL-Part images labeled as "Aeroplane".

Notably, employing a less precise set of concepts can introduce disturbances, as shown in the observations. As indicated in Section 3.2.2, the ambiguity associated with certain concepts, such as "Multi-doored", can lead to bimodal distributions (an airplane having one, multiple, or no doors). In Figure 7, we show the histogram of the clip scores $z$ of the images that have the class "aeroplane". Compared to the histogram of less ambiguous cases (like in Figure 3), we observe that the histogram presents anomalies, especially around the mean.

Additionally, we performed a similar experiment with larger sets of concepts. We selected random subsets containing 10, 15, and 20 concepts from the PASCAL-Part set listed in Table 6. The outcomes are displayed in Figure 8. In this scenario, it becomes obvious that the Gaussian assumption is increasingly violated as the number of concepts grows. Indeed, as the number of concepts increases, the likelihood of encountering

ambiguous concepts in each sample significantly increases, which undermines the feasibility of modeling the data as an unimodal Gaussian distribution.

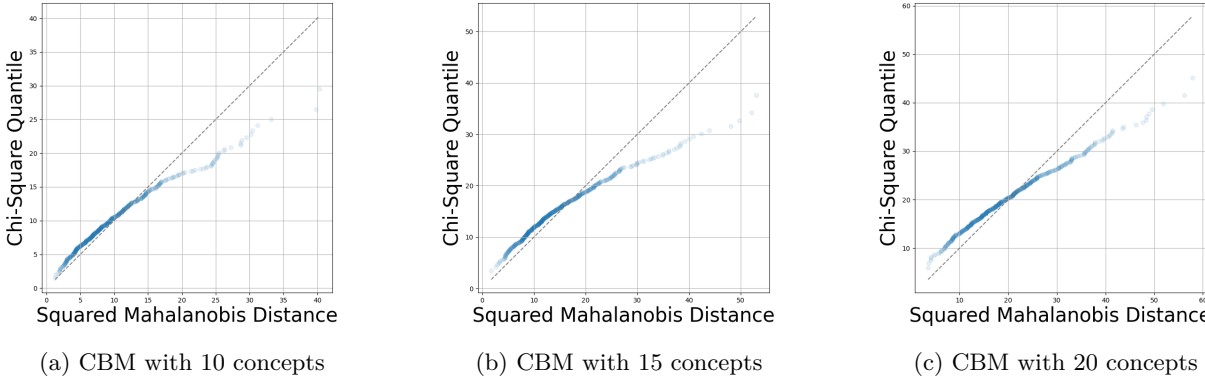

(a) CBM with 10 concepts (b) CBM with 15 concepts (c) CBM with 20 concepts

Figure 8: **Multivariate Q-Q plot illustrating the Gaussian fit for conditional modeling of the latent space, given a fixed label.** The plot is specifically for PASCAL-Part images labeled as "Aeroplane".

## 5.2 Assessing the Accuracy

**Comparison with other classifiers.** In this section, we undertake a comparative evaluation of the performance of our CLIP-QDA classifier in contrast to classifiers used in the baseline (see Section 4.2). Results of these experiments are available in Table 2. The image encoder used in all our experiments based on CLIP-based CBMs is $ViT - L/14@336px$ provided by OpenAI's public repository.

Table 2: **Test set accuracy.** On the top are methods that require full training on images, and on the bottom, CBMs. Because Greybox XAI, X-NeSyL and Morales Rodríguez et al. (2024) require concept annotations, their results on MIT scenes and ImageNet are not available.

| | Method | PASCAL-Part ↑ | MIT scenes ↑ | MonuMAI ↑ | ImageNet ↑ |
|---|---|---|---|---|---|
| **Non XAI** | Resnet 50 (He et al., 2016) | 0.84 | 0.86 | 0.95 | 0.80 |
| | ViT-L 336px (Dosovitskiy et al., 2021) | 0.95 | 0.94 | 0.94 | 0.85 |
| **CBMs** | Greybox XAI (Bennetot et al., 2022) | 0.88 | - | 0.94 | - |
| | X-NeSyL (Díaz-Rodríguez et al., 2022) | 0.82 | - | 0.90 | - |
| | Morales Rodríguez et al. (2024) | 0.86 | - | **0.98** | - |
| | CLIP (zero-shot) (Radford et al., 2021) | 0.81 | 0.63 | 0,52 | 0.76 |
| | LaBo (Yang et al., 2023) | 0.83 | 0.75 | 0.74 | 0.69 |
| | Yan et al. (2023b), Yan et al. (2023a) | **0.91** | 0.77 | 0.77 | **0.81** |
| | CLIP-QDA (ours) | 0.90 | **0.81** | 0.89 | 0.60 |

Our findings reveal that fine-tuning using CBM, either as a linear or a QDA probe, significantly improves performance, as evidenced by the increase in accuracy compared to using CLIP as a zero-shot classifier. This improvement is particularly pronounced on datasets dedicated to specialized tasks, such as MonuMAI. Additionally, CLIP CBMs tend to achieve performances comparable to networks trained from raw images, making these models appealing for image classification due to their reduced training cost in both time and resources, as well as their interpretability. Notably, CLIP-QDA demonstrates competitive performance, when compared to linear probe techniques.

However, CLIP-QDA faces challenges in delivering competitive results for datasets like PASCAL-Part and ImageNet, which feature a significantly larger number of labels and concepts. The accuracy decline could be attributed to the use of a broader set of concepts tailored specifically for these datasets. This challenges the Gaussian assumption and potentially impacts the effectiveness of our classifier. Notably, there appears to be a correlation between the tests conducted on the different datasets and the observations depicted in Figure 8. For instance, while MonuMAI uses 20 concepts, MIT scenes use 25, PASCAL-Part employs 80, and ImageNet involves a staggering 5000 concepts.

**Influence of the number of concepts.** To assess the influence of the number of concepts $C$ on the accuracy, we conduct experiments on the PASCAL-Part dataset. These experiments involve testing accuracy for both QDA and linear probe with concept sets of different lengths, all generated following the methodology described in Section A.3. As seen in Figure 9, CLIP-QDAperforms better than linear probe when the number of concepts is relatively low. In contrast, the linear probe outperforms CLIP-QDA as the number of concepts increases. This observation aligns with the insights gained from the discussion on Gaussian modeling in Section 5.1, where a higher number of concepts challenges the grounding assumptions of CLIP-QDA.

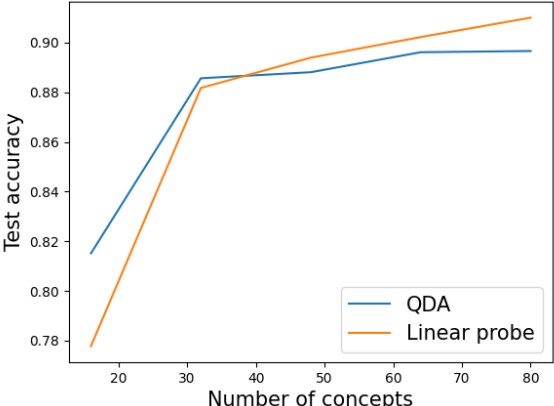

Figure 9: **Accuracy of the classifiers on PASCAL-Part for different concept sizes.** QDA refers to the use of CLIP-QDA as a classifier. Linear probe refers to the use of a linear layer as a classifier.

### 5.3  Assessing the Sample-wise Interpretability of XAI Methods

The objective of this subsection is to present our method within the context of current XAI methods (see Section 4.2) applied to CBMs. A comparative overview of the distinctive features of these methods is presented in Table 3. Since Greybox XAI, X-NeSyL and Morales Rodríguez et al. (2024) require additional annotations, we chose to omit these methods from our study.

Table 3: **Comparison of the different features of the XAI methods.** Top: existing methods. Bottom: ours. *Dataset-wise* refers to methods that provide dataset-wise explanations. *Sample-wise* refers to methods that provide sample-wise explanations. *Closed-form solution* refers to methods that do not require an optimization process to produce explanations. *Computable from weights* refers to methods that produce explanations from the model parameters and input values. *Image level* refers to methods that produce explanations from the image input. *Concept level* refers to methods that produce explanations from the concept input. *Compatible with CLIP-QDA* refers to methods that can produce explanations with CLIP-QDA as a classifier. *Compatible with CLIP linear probe* refers to methods that can produce explanations with CLIP linear probe as a classifier.

| Metric | Dataset wise | Sample wise | Closed-form solution | Computable from weights | Image level | Concept level | Compatible with CLIP-QDA | Compatible with CLIP linear probe |
|---|---|---|---|---|---|---|---|---|
| GradCAM | | ✓ | | | ✓ | | ✓ | ✓ |
| LIME | | ✓ | | | ✓ | | ✓ | ✓ |
| SHAP | | ✓ | | | ✓ | | ✓ | ✓ |
| LaBo | ✓ | | ✓ | ✓ | | ✓ | | |
| Yan et al | ✓ | ✓ | ✓ | ✓ | | ✓ | | ✓ |
| Greybox XAI | | ✓ | | | ✓ | ✓ | | |
| X-NeSyL | | ✓ | | | ✓ | ✓ | | |
| Morales et al | | ✓ | | | | ✓ | | |
| CLIP-LIME | | ✓ | | | | ✓ | ✓ | ✓ |
| CLIP-SHAP | | ✓ | | | | ✓ | ✓ | ✓ |
| QDA-CBM | ✓ | ✓ | ✓ | ✓ | | ✓ | ✓ | |

**Qualitative analysis.** To facilitate the comparison of various explanations generated for the same sample in the dataset, we computed the results for one image from PASCAL-Part (Donadello & Serafini, 2016). The results are categorized into two sections: image-level explanations and concept-level explanations. The subsequent section focuses on explaining the model's prediction process given the image 10a as the input, which is labeled as "person". The top two predicted labels by the classifier for this image are "person" and "potted plant". Note that additional samples are available in Section A.7.

Image-level explanations are presented in Figure 10. For each of these samples, a weight is assigned to each pixel. All the results are computed by applying these post-hoc explanation methods on CLIP-QDA. Additional implementation and computation details can be found in A.1. While the results are easy to comprehend, criticism may arise due to potential misunderstandings regarding the specific focus of the method. While these methods successfully highlight the object of interest, it is challenging to discern the exact pattern that the method emphasizes. Another observation is that, compared to classical classifiers, the results on CLIP-based CBMs are more imprecise, highlighting large areas of interest.

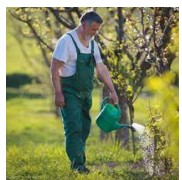 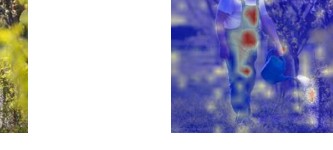 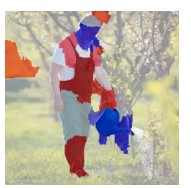 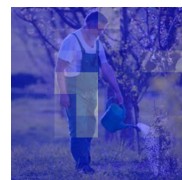

(a) **Input image**    (b) **GradCAM Explanation**    (c) **LIME Explanation**    (d) **SHAP explanation**

Figure 10: **Sample-wise explanations (image level).** Note that the classifier classified the image correctly.

Concept-level sample explanations are computed in Figure 11. On a sample-wise basis, CLIP-LIME and CLIP-SHAP are computed using CLIP-QDA. Given that the method is model-specific, the Yan et al. method is computed on a linear classifier.

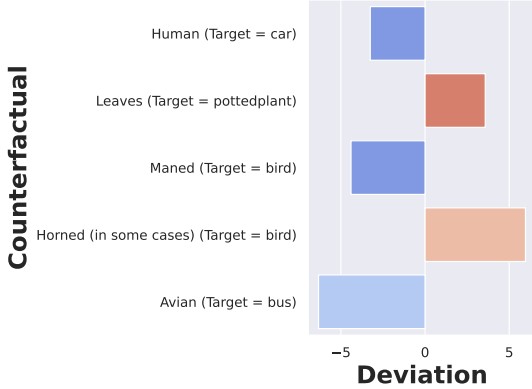

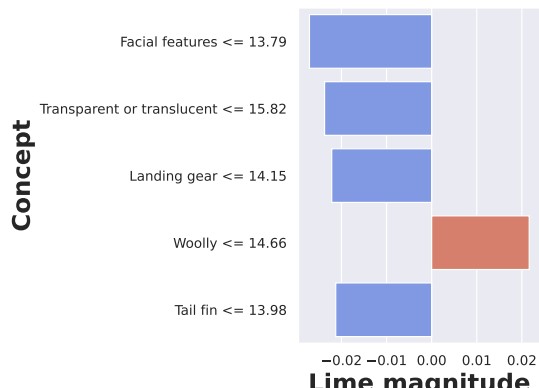

(a) **CLIP-QDA$^{local}$ explanation.** The first row can be read as follows: removing a little of the concept "Human" to the vector $z$ would change the label to "Car".

(b) **CLIP-LIME explanation.** The first row can be read as follows: the fact that the concept score of "Facial features" is below 13.79 has a negative impact on the predicted label.

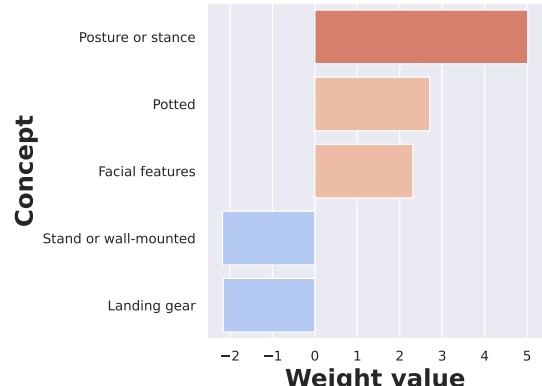

(c) **Yan et al. explanation (sample).** The first row can be read as follows: the concept "Posture or stance" has a positive impact on the predicted label.

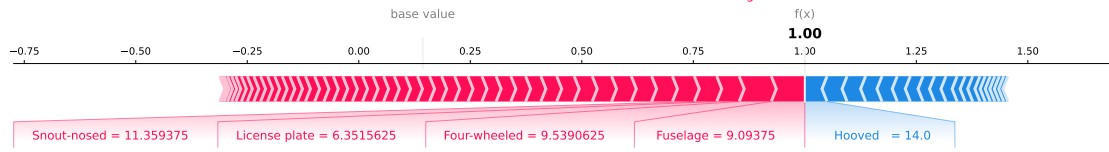

(d) **CLIP-SHAP explanation.** The highest value can be read as follows: the concept "Fuselage" has a positive impact on the predicted label.

Figure 11: **Sample-wise explanations (concept level) of the image 10a.** Except for CLIP-SHAP, only the top 5 concepts are displayed. Note that the classifier correctly labeled the image.

First, we notice that there is a high diversity of concepts displayed among the methods. Overall, there are concepts associated with the prediction label ("Posture or stance", "Human") and concepts that arise from the specificities of the image ("Leaves"). Some concepts seem irrelevant ("Landing Gear", "Windshield", "Wooly"), which may be explained by the fact that their scores are useful to the classification process despite their absence in the image. In addition, it is worth noting that the context provided by the counterfactual analysis helps to unveil insights into the reasons behind the explanation. For example, when taking into account the concept "Leaves" seems inappropriate, CLIP-QDA$^{local}$ explanations suggest that adding that concept influences the prediction of the class "potted plant," indicating that this concept is important because it reveals the decision process behind predicting a person rather than a plant.

**Quantitative analysis.** To conduct a more comprehensive analysis beyond qualitative observations, we subject the various methods to quantitative evaluations. We apply the procedure outlined in Section 4.3 to the PASCAL-Part and Cats/Dogs/Cars datasets. In addition to CLIP-LIME, CLIP-SHAP, and QDA-CBM explanations, we also tested the explanations provided by LaBo and Yan et al. For the selection of $S_s$ in Equation 8, in the case of the PASCAL-Part dataset, we opted to designate the concepts to detect as the concepts that are associated with the two highest probabilities of the model's inference (refer to the Classes/Concepts association in Table 6). In the Cats/Dogs/Cars dataset, we chose to identify the concepts "Black" and "White" in the biased setup to spot potential biases. The resulting *Det* score for this dataset is calculated by considering $S_s = [Black, White]$ for classifiers trained with biased setups as described in 1. To reduce uncertainties, we do not only take the *Det* score for the set [Black Cats, White Cars] but also for all possible similar biased binary classification tasks ([Black Cats, White Cars], [Black Dogs, White Cars], etc.). Additionally, we incorporated inference time as a parameter in our experiments referring to the time taken to produce explanations inferences on the entire validation set. Results on PASCAL-Part are displayed in Table 5 and results on Cats/Dogs/Cars are displayed in Table 4.

Table 4: **Quantitative results for different XAI methods (Cats/Dogs/Cars dataset).** Top: existing methods. Bottom: ours. Del refers to the deletion score (7) on a random set of concepts (Set 1) and the set of concepts defined in 6 (Set 2). Det refers to the detection score (8). The explanations of the three upper methods being direct, they present no computation time.

| Method | Del (Set 1) ↓ | Del (Set 2) ↓ | Det ↑ | Inference time (im/s) ↓ |
|---|---|---|---|---|
| *Yan et al.* | 0.6253 | 0.7522 | 0.3183 | / |
| *LaBo* | 0.5971 | 0.6224 | 0.2140 | / |
| *Random* | 0.8278 | 0.7474 | 0.1171 | / |
| | | | | |
| CLIP-QDA$^{local}$ (ours) | 0.7609 | 0.5820 | 0.2724 | **3.01** |
| *CLIP-LIME* (ours) | 0.5397 | 0.5646 | **0.4042** | 76.69 |
| *CLIP-SHAP* (ours) | **0.4821** | **0.3831** | 0.3696 | 256.76 |

Regarding the Cats/Dogs/Cars dataset, our methods exhibit superior performance compared to the state of the art, emphasizing the reliability of CLIP-QDA on datasets with a low number of concepts. Particularly, QDA-CBM demonstrates significantly faster inference times than CLIP-LIME and CLIP-SHAP, albeit at the expense of slightly lower deletion and detection scores. On datasets with a higher number of concepts and classes, such as PASCAL-Part, existing methods maintain higher scores.

## 5.4 Assessing the Dataset-wise Interpretability of XAI Methods

### 5.4.1 Evaluation of CLIP-QDA$^{global}$ on the PascalPART dataset

Following the setup expressed in Section 5.3, for dataset-wise explanations (Figure 12), we present the explanations for the CLIP-QDA$^{global}$ method, computed on CLIP-QDA, the LaBo explanation, computed on the LaBo classifier, and the Yan et al. explanation, computed on a linear classifier. Additional samples are available in Section A.7

Table 5: **Quantitative results for different XAI methods (PASCAL-Part dataset).** Top: existing methods. Bottom: ours. Del refers to the deletion score (7) on a random set of concepts (Set 1) and the usual set of concepts defined in 6 (Set 2). Det refers to the detection score (8). The explanations of the three upper methods being direct, they present no computation time.

| Method | Del (Set 1) ↓ | Del (Set 2) ↓ | Det ↑ | Inference time (im/s)↓ |
|---|---|---|---|---|
| *Yan et al.* | 0.6968 | 0.8824 | **0.4157** | / |
| *LaBo* | **0.4446** | 0.8822 | 0.4213 | / |
| *Random* | 0.8452 | 0.8999 | 0.1258 | / |
| | | | | |
| CLIP-QDA$^{local}$ (ours) | 0.7983 | 0.8189 | 0.1564 | **2341.12** |
| *CLIP-LIME* (ours) | 0.7313 | 0.8511 | 0.0873 | 2857.28 |
| *CLIP-SHAP* (ours) | 0.5698 | **0.5510** | 0.2679 | 7207.58 |

We observe that the concepts obtained from all the dataset-wise methods are generally coherent, including concepts commonly associated with persons, such as "Posture or stance" or "Facial features". However, in contrast to the other two methods that impose positive magnitudes, the CLIP-QDA$^{global}$ explanation allows for negative values, providing insights into whether the impact is negative or positive. Additionally, our CLIP-QDA$^{global}$ method emphasizes the comparison between two classes of interest, enabling a detailed exploration of specific disparities, like here between "potted plant" and "person".

### 5.4.2 Evaluation of CLIP-QDA$^{global}$ on the Cats/Dogs/Cars dataset

We also present a toy example from our Cats/Dogs/Cars dataset by constructing a CBM consisting of the concepts Table 6, plus the concepts "Black" and "White". Next, we display the 10 most influential concepts according to our global metric (the top 10 concepts that have the highest Wasserstein distance) in both biased and unbiased setups. Results are presented in Figure 13.

We can observe that the concepts "Black" and "White" hold significantly higher importance in the biased setup, indicating that the classifier is likely to be biased about these concepts. This shows that our global explanation method has the potential for detecting biases in datasets (Tommasi et al., 2017). Additional explanation samples for various use cases are available in Appendix A.7.

Finally, we show an application of our local metric within the framework of our biased setup, as previously described. Subsequently, we feed an image of a white cat into our classifier (Figure 14). It is noteworthy that the image is misclassified as a car. Our local metric demonstrates sensitivity to the dataset's color bias, corroborating the warning issued by the global explanation.

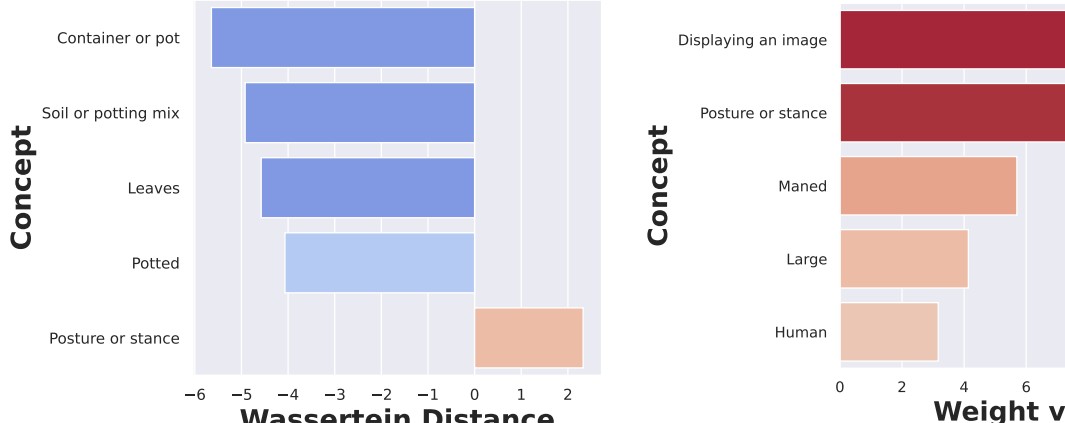

(a) **CLIP-QDA**$^{global}$ **explanation associated with the top two predictions ("person" and "potted plant").** The first row can be read as follows: the distribution of the concept "Container or pot" for the class "person" is mostly to the left (smaller values) of the one for the class "potted plant".

(b) **LaBo explanation associated with the top prediction ("person").** The first row can be read as follows: the weight associated with the concept "Displaying an image" has the highest value among the weights related to the class "person".

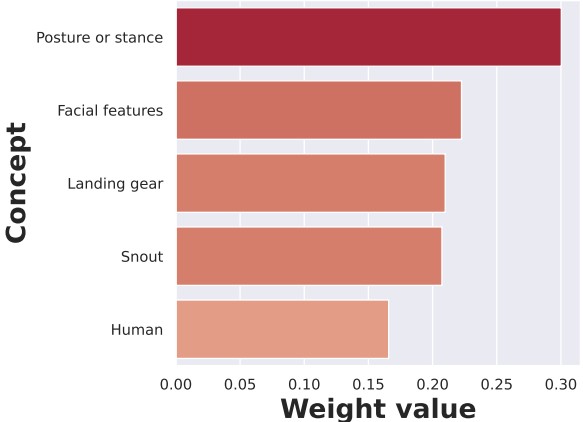

(c) **Yan et al. explanation (dataset) associated with the top prediction ("person").** The first row can be read as follows: the weight associated with the concept "Posture or stance" has the highest value among the weights related to the class "person".

Figure 12: **Dataset-wise explanations of the image 10a.** Only the top 5 concepts are displayed. Note that the classifier correctly labeled the image.

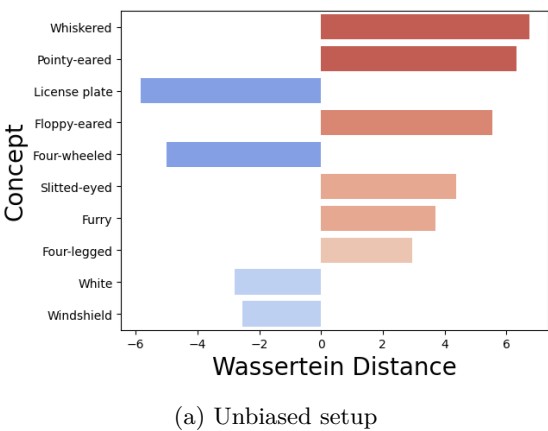

(a) Unbiased setup

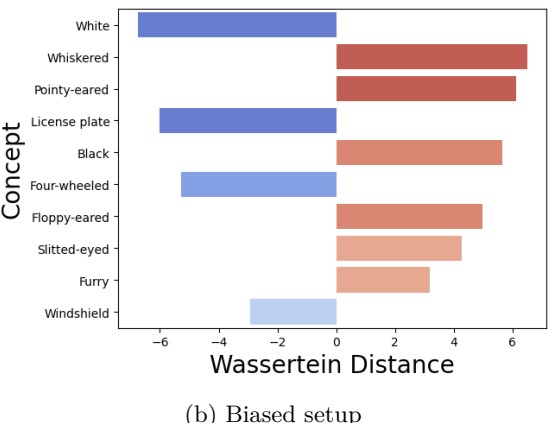

(b) Biased setup

Figure 13: **Global explanation on subsets of Cats/Dogs/Cars.** Here, $c_1$=“Cat” and $c_2$=“Car”. Positive values indicate concepts that are more prevalent in cat images than car images, while negative values indicate concepts that are more common in car images compared to cat images. We display here only the top 10 concepts that have the highest Wasserstein distance (the concept “Black” is positioned $15^{th}$ in the unbiased setup).

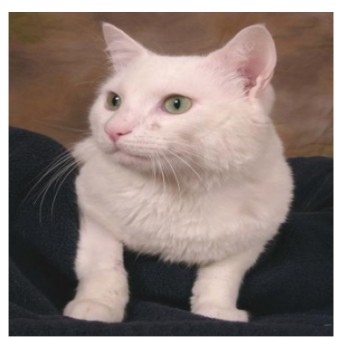

(a) Input image

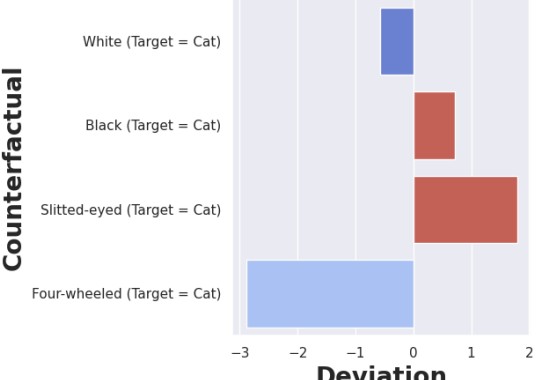

(b) **CLIP-QDA$^{local}$ explanation.** Only the top 5 concepts are displayed. Note that the classifier misclassified the image as “Car”. The first row must be read as follows: removing a little of the concept “White” to the vector $z$ induces a change of label to “Cat”.

Figure 14: **Local explanation on subsets of Cats/Dogs/Cars.** On the right figure, the blue/red scale represents the scaled counterfactuals as in equation 6, the text in each box corresponds to the label predicted after the perturbation, followed to the concept changed to obtain its result (in parentheses).

# 6 Conclusion

In this paper, we introduce a modeling approach for the embedding space of CLIP, a foundation model trained on a vast dataset. Our observations reveal that CLIP can organize information in a distribution that exhibits similarities with a mixture of Gaussians. Building upon this insight, we develop an adapted concept bottleneck model that demonstrates competitive performance along with transparency. While the model that we have presented offers the advantage of simplicity and a limited number of parameters, it does encounter challenges when dealing with a broader and more ambiguous set of concepts. As a suggestion for future research, we propose to extend this modeling approach to incorporate other priors, such as Laplacian distributions, and to explore more complex models, including those with multiple components, *i.e.*, using more than one Gaussian to describe a class for example. Another avenue to potentially enhance the performance of our model is to explore guiding the latent space of CLIP's scores towards a Gaussian distribution. Additionally, our research is centered around a specific embedding space (CLIP scores). Exploring similar work on other latent spaces, particularly those associated with multimodal foundation models, could be valuable to determine if similar patterns exist in those spaces.

# 7 Acknoledgements

This work was performed using HPC resources from GENCI-IDRIS (Grant 2023 - AD011014675).

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

# A Appendix

## A.1 Implementation details

**GradCAM.** To compute GradCAM explanations, we applied the method to the $21^{th}$ block of the image encoder in CLIP-QDA, using the PyTorch package *pytorch-grad-cam* (Gildenblat & contributors, 2021). Specifically, we generated the heatmap by backpropagating the gradient from the classifier's inference class, considering the entire network (CLIP+classifier).

**LIME.** The LIME image-level explanations are generated using the superpixel approach defined in (Ribeiro et al., 2016) using the official author's repository. All of our samples are generated using the default parameters for image-level explanations, i.e. an exponential kernel of width of 0.25 on an image segmented using quickshift clustering. Concerning the visualization, each explanation underlines the top 10 superpixels that exert the most influence on the prediction. Superpixels that positively contribute to the prediction of the label are in red and superpixels that positively contribute to the prediction of the label are in blue.

**SHAP.** The SHAP image-level explanations are generated using the partitioning approach defined in the official author's repository. All of our samples are generated using the default parameters for image-level explanations, i.e., running a blur image masker to partition the input image with rectangular masks. The estimation of Shapley values is obtained by running 200 evaluations. Partitions that positively contribute to the prediction of the label are in red and superpixels that negatively contribute to the prediction of the label are in blue.

**CLIP-LIME.** The CLIP-LIME explanations are generated using the approach used for tabular data as defined in (Ribeiro et al., 2016) using the official author's repository. All of our samples are generated using the default parameters for tabular-level explanations, i.e., an exponential kernel of width $0.75 \times N$, ($N$ being the number of concepts).

**CLIP-SHAP.** The CLIP-SHAP explanations are generated using the approach used for tabular data as in the official author's repository. All of our samples are generated using the default parameters for explanations, i.e., DeepLIFT algorithm using the training data as a background dataset.

**LaBo.** For LaBo, we train a LaBo classifier by using Adam optimizer with a learning rate of 0.5, a weight decay of 0, and a batch size of 8192. The resulting explanations follow the resulting weight matrix.

**Yan et al.** For the Yan et al. method, we train a linear classifier by using Adam optimizer with a learning rate of $5 \times 10^{-3}$, a weight decay of $10^{-4}$, and a batch size of 512. The sample-wise (concept level) explanation results from the product between the concept score and its associated weight. The dataset-wise explanation results from the weights alone.

**Resnet.** To train the Resnet classifier on PASCAL-Part, MIT scenes and MonuMAI, we initialized the network with ImageNet pertaining. Then, we trained the network using Adam optimizer with a learning rate of $10^{-3}$ for the probe, $10^{-4}$ for the backbone, a batch size of 64, a momentum of 0.9 and a weight decay of $10^{-4}$. For ImageNet results, we use results provided by the authors.

**ViT.** To train the ViT classifier on PASCAL-Part, MIT scenes and MonuMAI, we initialized the network with ImageNet pertaining. Then, we trained the network using Adam optimizer with a learning rate of $10^{-2}$ for the probe, a frozen backbone, a batch size of 128, a momentum of 0.9, and no weight decay. For ImageNet results, we use results provided by the authors.

### A.2 Cats/Dogs/Cars

The Cats/Dogs/Cars dataset is available on hugging face. To create this dataset, we combined two well-known datasets: the Kaggle Cats and Dogs Dataset (Cukierski, 2013) and the Standford Cars Dataset (Krause et al., 2013). We then filtered the dataset to include only images featuring black and white animals and cars, leading to six different categories. The primary objective behind assembling this dataset is to facilitate research in scenarios characterized by significant data bias, such as the classification of white cats when the training data predominantly consists of images of white dogs and black cats.

### A.3 Set of concepts

Inspired by Yang et al. (2023), we use large language models to provide concept sets. Concretely, we use GPT-3 (Brown et al., 2020) with the following preprompt: "In this task, you have to give visual descriptions

that describe an image. Respond as a list. Each item being a word." Then, we generate the sets of words by the following prompt: "What are [N] useful visual descriptors to distinguish a [class] in a photo?". By doing so, we generated 5 concepts per class, presented in Table 6. An additional ordering of concepts by subcategories is available in Table 7.

Table 6: List of concepts used, ordered by classes of interest.

| Dataset | Set of concepts |
| --- | --- |
| *PASCAL-Part* | 'aeroplane':[Winged, Jet engines, Tail fin, Fuselage, Landing gear],'bicycle':[Two-wheeled, Pedals, Handlebars, Frame, Chain-driven],'bird':[Feathery, Beaked, Wingspread, Perched, Avian],'bottle':[Glass or plastic, Cylindrical, Necked, Cap or cork, Transparent or translucent],'bus':[Large, Rectangular, Windows, Wheels, Multi-doored],'cat':[Furry, Whiskered, Pointy-eared, Slitted-eyed, Four-legged],'car':[Metallic, Four-wheeled, Headlights, Windshield, License plate],'dog':[Snout, Wagging-tailed, Snout-nosed, Floppy-eared, Tail-wagging],'cow':[Bovine, Hooved, Horned (in some cases), Spotted or solid-colored, Grazing (if in a field)],'horse':[Equine, Hooved, Maned, Tailed, Galloping (if in motion)],'motorbike':[Two-wheeled , Engine, Handlebars , Exhaust, Saddle or seat],'person':[Human, Facial features, Limbs (arms and legs), Clothing, Posture or stance],'potted plant':[Potted, Green, Leaves, Soil or potting mix, Container or pot],'sheep':[Woolly, Hooved, Grazing, Herded (if in a group), White or colored fleece],'train':[Locomotive, Railcars, Tracks, Wheels , Carriages],'tvmonitor':[Screen, Rectangular , Frame or bezel, Stand or wall-mounted, Displaying an image] |
| *MonuMAI* | 'Baroque':[Ornate, Elaborate sculptures, Intricate details, Curved or asymmetrical design, Historical or aged appearance],'Gothic':[Pointed arches, Ribbed vaults, Flying buttresses, Stained glass windows, Tall spires or towers],'Hispanic muslim':[Mudejar style, Intricate geometric patterns, Horseshoe arches, Decorative tilework (azulejos), Islamic-inspired motifs],'Rennaissance':[Classical proportions, Symmetrical design, Columns and pilasters, Human statues and sculptures, Dome or dome-like structures] |
| *MIT scenes* | 'Store':[Building or structure, Signage or banners, Glass windows or doors, Displayed products or merchandise, People entering or exiting (if applicable)],'Home':[Residential, Roofed, Windows, Landscaping or yard, Front entrance or door],'Public space':[Open area, Crowds (if people are present), Benches or seating, Pathways or walkways, Architectural features (e.g., buildings, statues)],'Leisure':[Recreational, Play equipment (if applicable), Greenery or landscaping, Picnic tables or seating, Relaxing atmosphere],'Working space':[Office equipment (e.g., desks, computers), Task-oriented, Office chairs, Organized or structured, People working (if applicable)] |
| *Cats/Dogs/Cars* | 'Cat':[Furry, Whiskered, Pointy-eared, Slitted-eyed, Four-legged],'Car':[Metallic, Four-wheeled, Headlights, Windshield, License plate],'Dog':[Snout, Wagging-tailed, Snout-nosed, Floppy-eared, Tail-wagging] |

Table 7: List of concepts used, ordered by subcategories.

| Dataset | Set of concepts |
|---------|-----------------|
| *PASCAL-Part* | 'Objects':[Screen, Rectangular, Frame or bezel, Stand or wall-mounted, Displaying an image, Glass or plastic, Cylindrical, Necked, Cap or cork, Transparent or translucent, Potted, Green, Leaves, Soil or potting mix, Container or pot], 'Transportation-related':[Two-wheeled, Engine, Handlebars, Exhaust, Saddle or seat, Four-wheeled, Headlights, Windshield, License plate, Wheels, Multi-doored, Locomotive, Railcars, Tracks, Carriages, Chain-driven],'Aircraft-related':[Winged, Jet engines, Tail fin, Fuselage, Landing gear],'Building/Structure-related':[Large, Rectangular, Windows] 'Human Characteristics':[Human, Facial features, Limbs (arms and legs), Clothing, Posture or stance],'Avian Characteristics':[Feathery, Beaked, Wingspread, Perched, Avian],'Animal Characteristics':[Furry, Whiskered, Pointy-eared, Slitted-eyed, Four-legged, Equine, Hooved, Maned, Tailed, Galloping (if in motion), Snout, Wagging-tailed, Snout-nosed, Floppy-eared, Tail-wagging, Woolly, Grazing, Herded (if in a group), White or colored fleece, Bovine, Horned (in some cases), Spotted or solid-colored] |
| *MonuMAI* | 'Architectural Styles and Elements':[Mudejar style, Intricate geometric patterns, Horseshoe arches, Decorative tilework (azulejos), Islamic-inspired motifs, Stained glass windows, Pointed arches, Ribbed vaults],'Artistic Details and Features':[Ornate, Elaborate sculptures, Intricate details, Curved or asymmetrical design, Human statues and sculptures, Historical or aged appearance],'Architectural Components':[Flying buttresses, Classical proportions, Symmetrical design, Columns and pilasters, Dome or dome-like structures, Tall spires or towers] |
| *MIT scenes* | 'Work Environment':[Office equipment (e.g., desks, computers), Task-oriented, Office chairs, Organized or structured, People working (if applicable)], 'Leisure Environment':[Play equipment (if applicable), Greenery or landscaping, Picnic tables or seating, Relaxing atmosphere, Recreational],'Community Environment':[Roofed, Windows, Landscaping or yard, Front entrance or door, Building or structure, Signage or banners, Glass windows or doors, Displayed products or merchandise, People entering or exiting (if applicable), Open area, Crowds (if people are present), Benches or seating, Pathways or walkways, Architectural features (e.g., buildings, statues), Residential] |
| *Cats/Dogs/Cars* | 'Vehicle Features':[License plate, Headlights, Windshield, Four-wheeled], 'Animal Features':[Wagging-tailed, Tail-wagging, Snout, Whiskered, Pointy-eared, Slitted-eyed, Floppy-eared, Furry, Snout-nosed, Four-legged],'Material/Texture':[Metallic] |

## A.4  Study of images according to their position in the latent space

In addition to the CLIP-QDA$^{global}$ method, we introduce another approach leveraging multivariate Gaussian modeling to generate dataset-wise visual explanations for our model. Specifically, for a given class $c$, we compute the Mahalanobis distance with respect to $\mathcal{N}(\boldsymbol{z} \mid \boldsymbol{\mu}_c, \boldsymbol{\Sigma}_c)$ for each sample's latent space projection $\boldsymbol{z}$ labeled as $c$ within a dataset of interest:

$$d_M(\boldsymbol{z}, \boldsymbol{\mu}_c, \boldsymbol{\Sigma}_c) = \sqrt{(\boldsymbol{z} - \boldsymbol{\mu}_c)^{\top} \boldsymbol{\Sigma}_c^{-1} (\boldsymbol{z} - \boldsymbol{\mu}_c)}. \tag{9}$$

Then, we plot in Figures 15, 16, 17 and 18, the closest and farthest images with the dataset, according to this distance for PascalPART and MonuMAI.

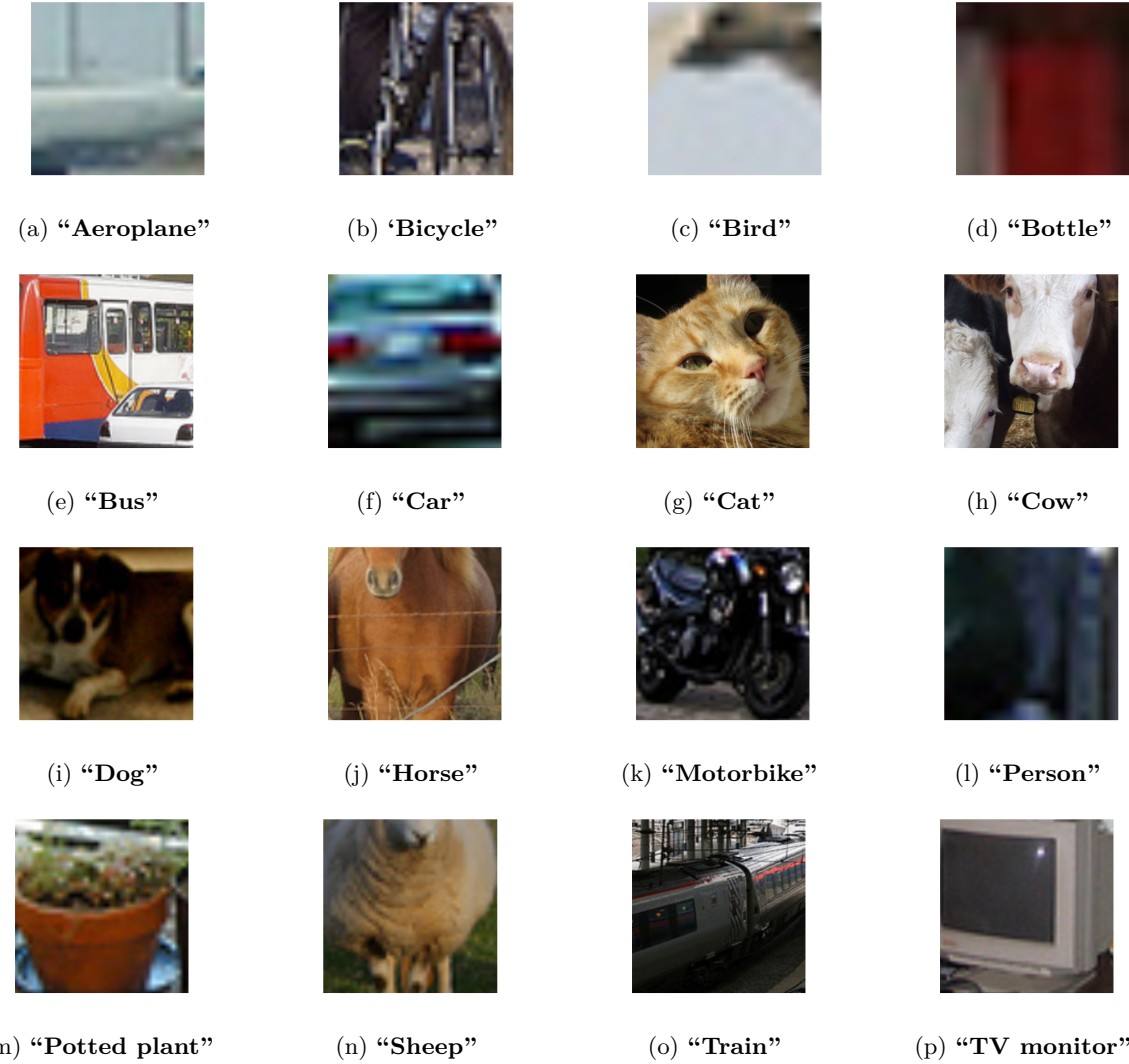

(a) **"Aeroplane"**    (b) **'Bicycle"**    (c) **"Bird"**    (d) **"Bottle"**

(e) **"Bus"**    (f) **"Car"**    (g) **"Cat"**    (h) **"Cow"**

(i) **"Dog"**    (j) **"Horse"**    (k) **"Motorbike"**    (l) **"Person"**

(m) **"Potted plant"**    (n) **"Sheep"**    (o) **"Train"**    (p) **"TV monitor"**

Figure 15: **Images of minimal distance for each class of PascalPART.** Each subfigure in the plot corresponds to the image of class $c$ from the test set that exhibits the minimal Mahalanobis distance, as defined in Equation 9, with respect to the Gaussian representation of the given class $c$.

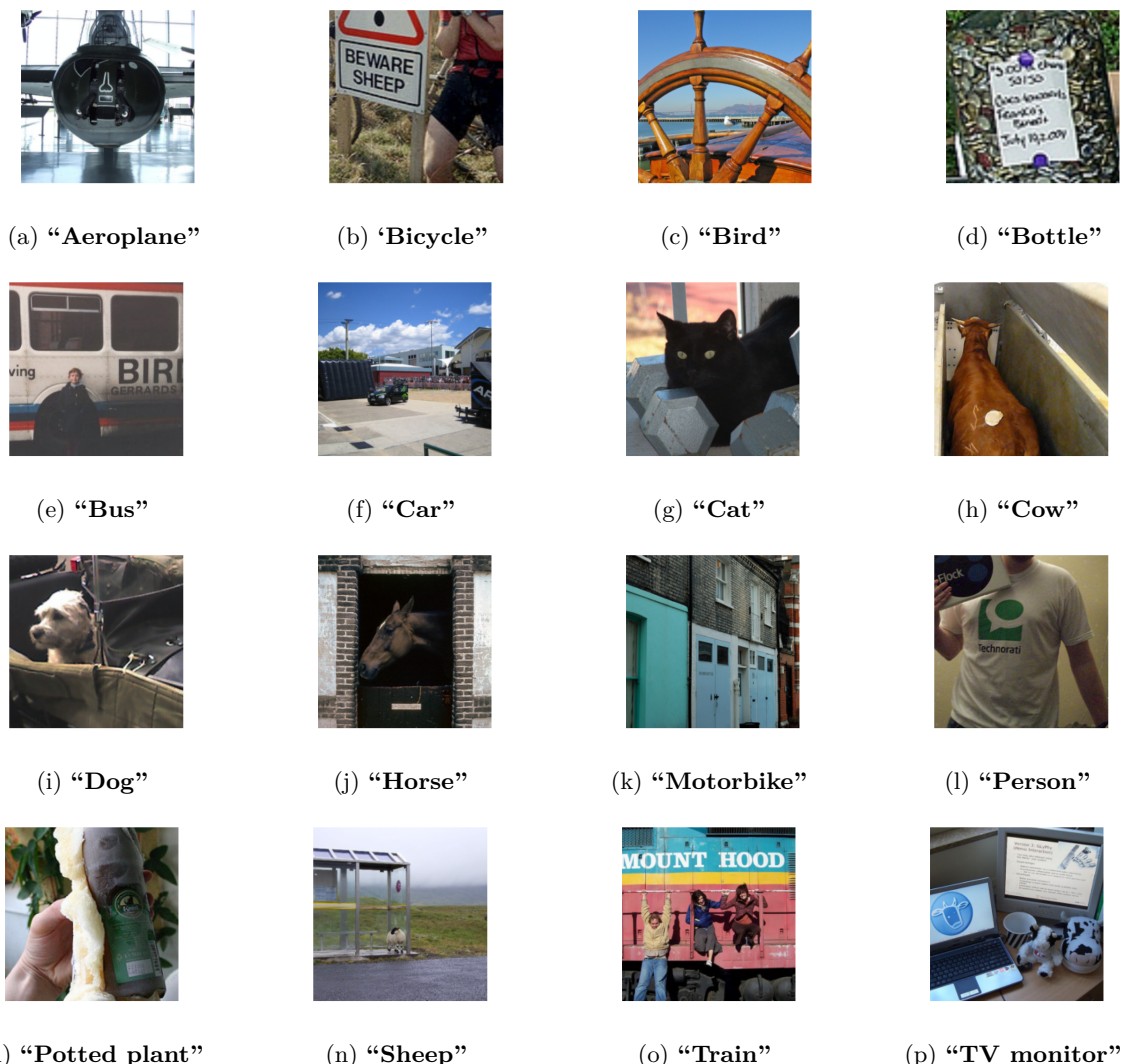

(a) **"Aeroplane"**   (b) **'Bicycle"**   (c) **"Bird"**   (d) **"Bottle"**

(e) **"Bus"**   (f) **"Car"**   (g) **"Cat"**   (h) **"Cow"**

(i) **"Dog"**   (j) **"Horse"**   (k) **"Motorbike"**   (l) **"Person"**

(m) **"Potted plant"**   (n) **"Sheep"**   (o) **"Train"**   (p) **"TV monitor"**

Figure 16: **Images of maximal distance for each class of PascalPART.** Each subfigure in the plot corresponds to the image of class $c$ from the test set that exhibits the maximal Mahalanobis distance, as defined in Equation 9, with respect to the Gaussian representation of the given class $c$.

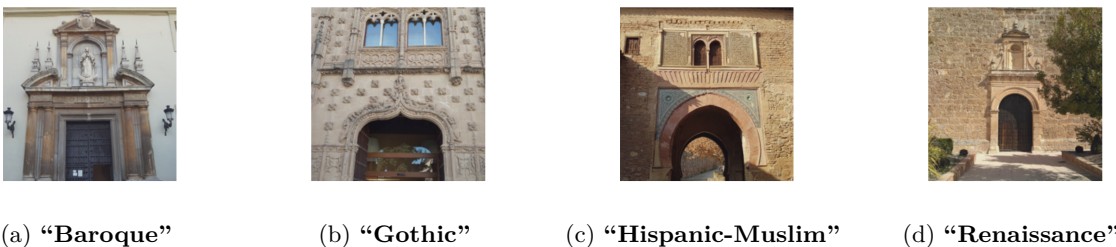

(a) **"Baroque"**   (b) **"Gothic"**   (c) **"Hispanic-Muslim"**   (d) **"Renaissance"**

Figure 17: **Images of minimal distance for each class of MonuMAI.** Each subfigure in the plot corresponds to the image of class $c$ from the test set that exhibits the minimal Mahalanobis distance, as defined in Equation 9, with respect to the Gaussian representation of the given class $c$.

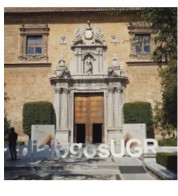
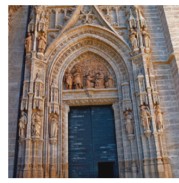
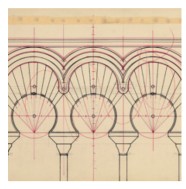
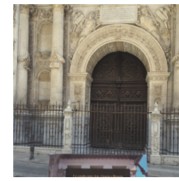

(a) **"Baroque"**  (b) **'Gothic"**  (c) **"Hispanic-Muslim"**  (d) **"Renaissance"**

Figure 18: **Images of maximal distance for each class of MonuMAI.** Each subfigure in the plot corresponds to the image of class $c$ from the test set that exhibits the minimal Mahalanobis distance, as defined in Equation 9, with respect to the Gaussian representation of the given class $c$.

For PascalPART, the closest images tend to be predominantly close-up shots of the object in question. However, this occasionally presents challenges due to the inherent pixelation of the images, leading to somewhat counterintuitive results. Conversely, the farthest images are typically those where the object is distant. Another scenario of failure arises when there are two classes present within the image, which complicates the retrieval process. On the other hand, for MonumAI, the closest images appear to be the simplest ones, without sign of disturbance. In contrast, the farthest images may include extraneous objects, such as signs.

### A.5 Closed-form solution of counterfactuals for CLIP QDA

First, we will derive the resolution of equation 5 for the binary case before extending it to the multiclass case. Let us begin with a binary classifier, where $Y \in \{h_{\boldsymbol{\omega}^h}(\boldsymbol{z}), \overline{h_{\boldsymbol{\omega}^h}(\boldsymbol{z})}\}$ (for convenience, we note $h_{\boldsymbol{\omega}^h}(\boldsymbol{z}) = c_h$ and $\overline{h_{\boldsymbol{\omega}^h}(\boldsymbol{z})} = c_{\overline{h}}$ ).

**Proposition 2.** *Let $h_{\boldsymbol{\omega}^h}$ a binary classifier and $\boldsymbol{Z}$ following a mixture of Gaussians such as $\boldsymbol{Z}_{Y=c_h} \sim \mathcal{N}(\boldsymbol{\mu}_{c_h}, \boldsymbol{\Sigma}_{c_h})$ and $\boldsymbol{Z}_{Y=c_{\overline{h}}} \sim \mathcal{N}(\boldsymbol{\mu}_{c_{\overline{h}}}, \boldsymbol{\Sigma}_{c_{\overline{h}}}^{-1})$, and $\boldsymbol{\epsilon}_s^j$ a perturbation with the above sparsity and sign restrictions. Then, there is a closed form solution to problem 5, given by:*

$$
\epsilon_s^j = \begin{cases}
\emptyset & \text{if } b^2 - 4Pc < 0 \\
& \text{or } (s \neq \text{sign}(b_1) \text{ and } s \neq \text{sign}(b_2)) \\
\\
b_1 & \text{if } b^2 - 4Pc > 0 \text{ and } \text{sign}(b_1) = s \\
& \text{and } (\text{sign}(b_2) \neq s \text{ or } |b_2| \geq |b_1|) \\
\\
b_2 & \text{if } b^2 - 4Pc > 0 \text{ and } \text{sign}(b_2) = s \\
& \text{and } (\text{sign}(b_1) \neq s \text{ or } |b_1| > |b_2|),
\end{cases}
\tag{10}
$$

*with:*

$$
P = \frac{1}{2}\left([\boldsymbol{\Sigma}_{c_{\overline{h}}}^{-1}]_{(j,j)} - [\boldsymbol{\Sigma}_{c_h}^{-1}]_{(j,j)}\right)
$$

$$
b = \sum_{k=1}^{N}\left(([\boldsymbol{z}]_{(k)} - [\boldsymbol{\mu}_{c_{\overline{h}}}]_{(k)})[\boldsymbol{\Sigma}_{c_{\overline{h}}}^{-1}]_{(j,k)} - ([\boldsymbol{z}]_{(k)} - [\boldsymbol{\mu}_{c_h}]_{(k)})[\boldsymbol{\Sigma}_{c_h}^{-1}]_{(j,k)}\right)
$$

$$
c = \frac{1}{2}\log\left[\frac{|\boldsymbol{\Sigma}_{c_{\overline{h}}}|}{|\boldsymbol{\Sigma}_{c_h}|}\right] + \log\left[\frac{p_{c_h}}{p_{c_{\overline{h}}}}\right] + \frac{1}{2}(\boldsymbol{z} - \boldsymbol{\mu}_{c_{\overline{h}}})^{\top}\boldsymbol{\Sigma}_{c_{\overline{h}}}^{-1}(\boldsymbol{z} - \boldsymbol{\mu}_{c_{\overline{h}}}) - \frac{1}{2}(\boldsymbol{z} - \boldsymbol{\mu}_{c_h})^{\top}\boldsymbol{\Sigma}_{c_h}^{-1}(\boldsymbol{z} - \boldsymbol{\mu}_{c_h})
$$

$$
b_1 = \frac{-b - \sqrt{b^2 - 4Pc}}{2P}
\tag{11}
$$

$$
b_2 = \frac{-b + \sqrt{b^2 - 4Pc}}{2P}.
$$

*Proof.* For the binary case, equation 5 can be written as :

$$\min \|\boldsymbol{\epsilon}_s^j\|^2 \quad \text{s.t.} \quad \frac{p_{c_{\overline{h}}}}{(2\pi)^{N/2}|\boldsymbol{\Sigma}_{c_{\overline{h}}}|^{\frac{1}{2}}} e^{-\frac{1}{2}(\boldsymbol{z}+\boldsymbol{\epsilon}_s^j-\boldsymbol{\mu}_{c_{\overline{h}}})^T \boldsymbol{\Sigma}_{c_{\overline{h}}}^{-1}(\boldsymbol{z}+\boldsymbol{\epsilon}_s^j-\boldsymbol{\mu}_{c_{\overline{h}}})} \leqslant \frac{p_{c_h}}{(2\pi)^{N/2}|\boldsymbol{\Sigma}_{c_h}|^{\frac{1}{2}}} e^{-\frac{1}{2}(\boldsymbol{z}+\boldsymbol{\epsilon}_s^j-\boldsymbol{\mu}_{c_h})^T \boldsymbol{\Sigma}_{c_h}^{-1}(\boldsymbol{z}+\boldsymbol{\epsilon}_s^j-\boldsymbol{\mu}_{c_h})}.$$

$$(12)$$

Let us focus on the inequality constraint:

$$\frac{p_{c_{\overline{h}}}}{|\boldsymbol{\Sigma}_{c_{\overline{h}}}|^{\frac{1}{2}}} e^{-\frac{1}{2}(\boldsymbol{z}+\boldsymbol{\epsilon}_s^j-\boldsymbol{\mu}_{c_{\overline{h}}})^T \boldsymbol{\Sigma}_{c_{\overline{h}}}^{-1}(\boldsymbol{z}+\boldsymbol{\epsilon}_s^j-\boldsymbol{\mu}_{c_{\overline{h}}})} \leqslant \frac{p_{c_h}}{|\boldsymbol{\Sigma}_{c_h}|^{\frac{1}{2}}} e^{-\frac{1}{2}(\boldsymbol{z}+\boldsymbol{\epsilon}_s^j-\boldsymbol{\mu}_{c_h})^T \boldsymbol{\Sigma}_{c_h}^{-1}(\boldsymbol{z}+\boldsymbol{\epsilon}_s^j-\boldsymbol{\mu}_{c_h})}$$

$$\Longleftrightarrow \log(p_{c_{\overline{h}}}) - \frac{1}{2}\log|\boldsymbol{\Sigma}_{c_{\overline{h}}}| - \frac{1}{2}(\boldsymbol{z}+\boldsymbol{\epsilon}_s^j-\boldsymbol{\mu}_{c_{\overline{h}}})^T \boldsymbol{\Sigma}_{c_{\overline{h}}}^{-1}(\boldsymbol{z}+\boldsymbol{\epsilon}_s^j-\boldsymbol{\mu}_{c_{\overline{h}}})$$

$$\leqslant \log(p_{c_h}) - \frac{1}{2}\log|\boldsymbol{\Sigma}_{c_h}| - \frac{1}{2}(\boldsymbol{z}+\boldsymbol{\epsilon}_s^j-\boldsymbol{\mu}_{c_h})^T \boldsymbol{\Sigma}_{c_h}^{-1}(\boldsymbol{z}+\boldsymbol{\epsilon}_s^j-\boldsymbol{\mu}_{c_h})$$

$$\Longleftrightarrow \log(p_{c_{\overline{h}}}) - \frac{1}{2}\log|\boldsymbol{\Sigma}_{c_{\overline{h}}}| - \frac{1}{2}(\boldsymbol{z}-\boldsymbol{\mu}_{c_{\overline{h}}})^T \boldsymbol{\Sigma}_{c_{\overline{h}}}^{-1}(\boldsymbol{z}-\boldsymbol{\mu}_{c_{\overline{h}}}) - \frac{1}{2}\boldsymbol{\epsilon}_s^{jT}\boldsymbol{\Sigma}_{c_{\overline{h}}}^{-1}\boldsymbol{\epsilon}_s^j - \boldsymbol{\epsilon}_s^{jT}\boldsymbol{\Sigma}_{c_{\overline{h}}}^{-1}(\boldsymbol{z}-\boldsymbol{\mu}_{c_{\overline{h}}})$$

$$\leqslant \log(p_{c_h}) - \frac{1}{2}\log|\boldsymbol{\Sigma}_{c_h}| - \frac{1}{2}(\boldsymbol{z}-\boldsymbol{\mu}_{c_h})^T \boldsymbol{\Sigma}_{c_h}^{-1}(\boldsymbol{z}-\boldsymbol{\mu}_{c_h}) - \frac{1}{2}\boldsymbol{\epsilon}_s^{jT}\boldsymbol{\Sigma}_{c_h}^{-1}\boldsymbol{\epsilon}_s^j - \boldsymbol{\epsilon}_s^{jT}\boldsymbol{\Sigma}_{c_h}^{-1}(\boldsymbol{z}-\boldsymbol{\mu}_{c_h})$$

$$\Longleftrightarrow \log(p_{c_{\overline{h}}}) - \frac{1}{2}\log|\boldsymbol{\Sigma}_{c_{\overline{h}}}| - \frac{1}{2}(\boldsymbol{z}-\boldsymbol{\mu}_{c_{\overline{h}}})^T \boldsymbol{\Sigma}_{c_{\overline{h}}}^{-1}(\boldsymbol{z}-\boldsymbol{\mu}_{c_{\overline{h}}}) - \frac{1}{2}[\boldsymbol{\Sigma}_{c_{\overline{h}}}^{-1}]_{(j,j)}(\epsilon_s^j)^2 - \epsilon_s^j \sum_{k=1}^{N}([\boldsymbol{z}]_{(k)} - [\boldsymbol{\mu}_{c_{\overline{h}}}]_{(k)})[\boldsymbol{\Sigma}_{c_{\overline{h}}}^{-1}]_{(j,k)}$$

$$\leqslant \log(p_{c_h}) - \frac{1}{2}\log|\boldsymbol{\Sigma}_{c_h}| - \frac{1}{2}(\boldsymbol{z}-\boldsymbol{\mu}_{c_h})^T \boldsymbol{\Sigma}_{c_h}^{-1}(\boldsymbol{z}-\boldsymbol{\mu}_{c_h}) - \frac{1}{2}[\boldsymbol{\Sigma}_{c_h}^{-1}]_{(j,j)}(\epsilon_s^j)^2 - \epsilon_s^j \sum_{k=1}^{N}([\boldsymbol{z}]_{(k)} - [\boldsymbol{\mu}_{c_h}]_{(k)})[\boldsymbol{\Sigma}_{c_h}^{-1}]_{(j,k)}$$

$$\Longleftrightarrow \qquad\qquad\qquad\qquad P(\epsilon_s^j)^2 + b\epsilon_s^j + c \geqslant 0.$$

Then we can rewrite the problem as :

$$\min \ (\boldsymbol{\epsilon}_s^j)^2 \quad \text{s.t.} \quad P(\epsilon_s^j)^2 + b\epsilon_s^j + c \leqslant 0. \tag{13}$$

To solve this problem, we introduce slack variables $\lambda$ and $I$, and define a Lagrangian as:

$$L(\boldsymbol{\epsilon}_s^j, \lambda, I) = (\epsilon_s^j)^2 + \lambda(P(\epsilon_s^j)^2 + b\epsilon_s^j + c + I^2). \tag{14}$$

Then, we can find the possible solutions by solving the system :

$$\begin{cases} \frac{\partial L}{\partial \epsilon_s^j} = 0 \\ \frac{\partial L}{\partial \lambda} = 0 \\ \frac{\partial L}{\partial I} = 0, \end{cases}$$

which corresponds to:

$$\begin{cases} 2(\lambda P + 1)\epsilon_s^j + \lambda b = 0 \\ P(\epsilon_s^j)^2 + b\epsilon_s^j + c + I^2 = 0 \\ 2\lambda I = 0. \end{cases} \tag{15}$$

The third equation of 15 indicates whether the inequality condition is saturated or not. If it is not saturated ($\lambda = 0$), then the label $c_{\overline{h}}$ is already achieved for $\boldsymbol{z}$, resulting in a solution of $\epsilon_{s,*}^j = 0$. This being impossible by construction, we only focus on the case where $\lambda \neq 0$.

If $\lambda \neq 0$, the condition is saturated, the second equation leads to:

$$P(\epsilon_s^j)^2 + b\epsilon_s^j + c = 0,$$

whose solutions are:

$$\epsilon_s^j \in \left\{ \frac{-b - \sqrt{b^2 - 4Pc}}{2P}, \frac{-b + \sqrt{b^2 - 4Pc}}{2P} \right\} \quad \text{if} \quad b^2 - 4Pc > 0,$$

which we rewrite as:

$$\epsilon_s^j \in \{b_1, b_2\} \quad \text{if} \quad b^2 - 4Pc > 0. \tag{16}$$

Considering 16, the validity of the results is conditioned by the sign $s$ and the condition of the magnitude of $\epsilon_{s,*}^j$. Then, the final result of the problem 5 is either $\emptyset$, $b_1$ or $b_2$ depending on the conditions:

$$
\begin{cases}
\emptyset \quad \text{if} \quad b^2 - 4Pc < 0 \\
\quad \text{or} \ (s \neq \text{sign}(b_1) \ \text{and} \ s \neq \text{sign}(b_2)) \\
\\
b_1 \quad \text{if} \quad b^2 - 4Pc > 0 \ \text{and} \ \text{sign}(b_1) = s \\
\quad \text{and} \ (\text{sign}(b_2) \neq s \ \text{or} \ |b_2| \geq |b_1|) \\
\\
b_2 \quad \text{if} \quad b^2 - 4Pc > 0 \ \text{and} \ \text{sign}(b_2) = s \\
\quad \text{and} \ (\text{sign}(b_1) \neq s \ \text{or} \ |b_1| > |b_2|).
\end{cases}
\tag{17}
$$

$\square$

To expand problem 5 to multiclass classification $C > 2$, we consider it as a succession of $C - 1$ binary classifications between each $i' \neq c_h$ and $c_h$. Given a concept index $j$ and a sign $s$, if we denote the solutions of theses problems as the set $\{\epsilon_{+,*,1}^j, ..., \epsilon_{s,*,c_h-1}^j, \epsilon_{s,*,c_h+1}^j, ..., \epsilon_{s,*,C}^j\}$, the final solution $\epsilon_{s,*}^j$ is if it exists, the value of minimal magnitude among this set.

Examples of explanations based on this metric are given in Sections 5.4 and A.7.

### A.6    Sample-wise explanation time

In Table 8, we display the amount of time taken to produce a sample-wise explanation for each image of the test set of PASCAL-Part and Cats/Dogs/Cars.

Table 8: Time (in seconds) to produce explanations on the entire test set.

| Method | CLIP-QDA$^{local}$ | CLIP-LIME | CLIP-SHAP |
|---|---|---|---|
| *Cats/Dogs/Cars* | 3.01 | 76.69 | 256.76 |
| *PASCAL-Part* | 2341.12 | 2857.28 | 7207.58 |
| *MonuMAI* | 2.19 | 19.94 | 56.25 |
| *MITscenes* | 33.41 | 509.33 | 1398.64 |

In this table, we can observe that using the local explanation is the fastest, especially for the cases where the number of concepts and classes are low. This is justified by the fact that our method consists of using Proposition 2 for each concept, sign, and class other than the inference one. Hence the complexity of this computation for each sample is $O(2(C - 1)N)$, where LIME's complexity does not depend on $C$.

### A.7    Additional samples

We display here additional samples of both sample-wise and dataset-wise explanations from the PASCAL-Part and Cats/Dogs/Cars dataset.

(a) Input image      (b) GradCAM Explanation    (c) LIME Explanation     (d) SHAP explanation

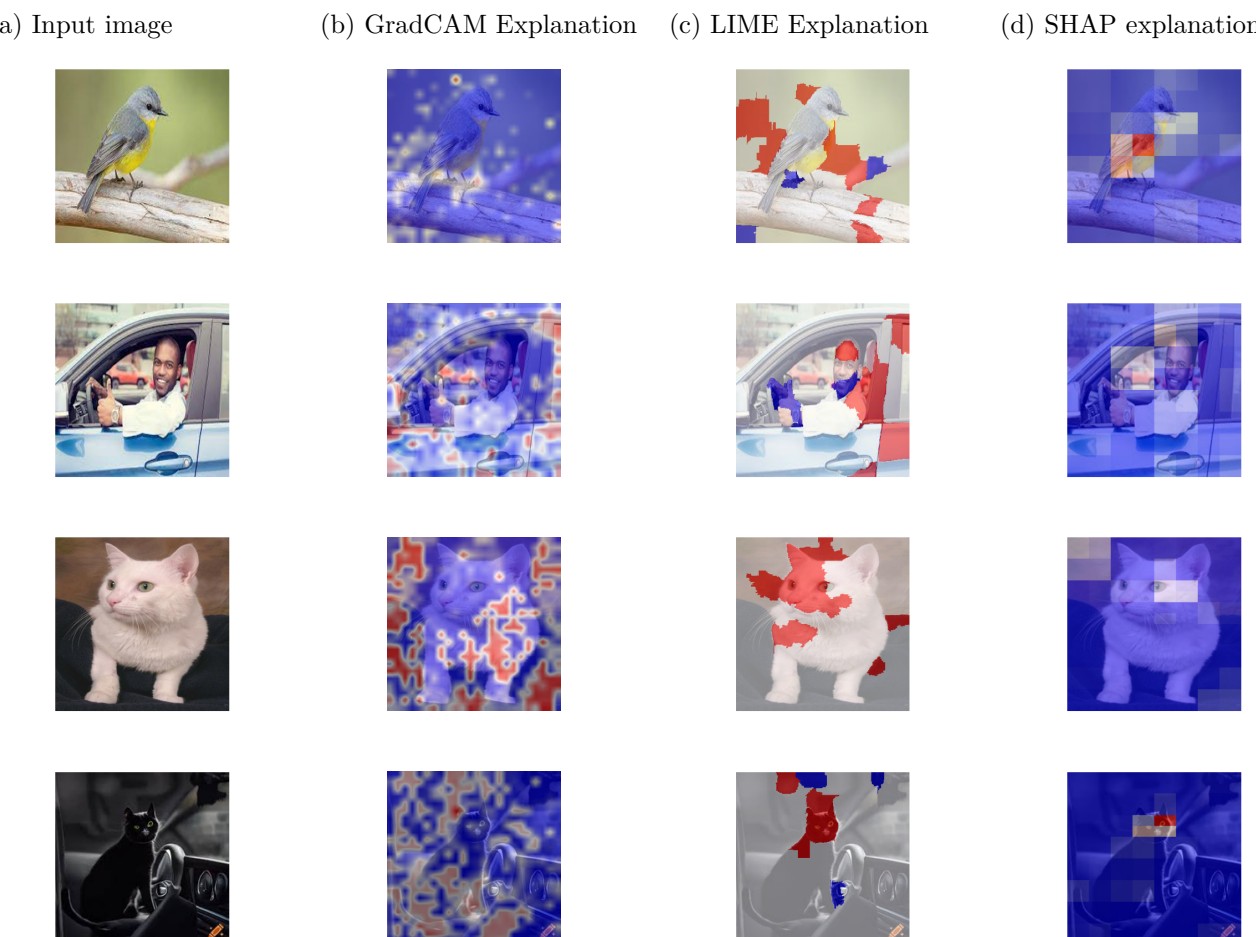

Figure 19: **Sample-wise explanations (image level).** The first two examples come from the PascalPART dataset, and the last two samples come from the Cats/Dogs/Cars dataset in the biased setup. Note that the classifier mislabeled the $2^{nd}$ example as "car" and the $4^{th}$ example as "car".

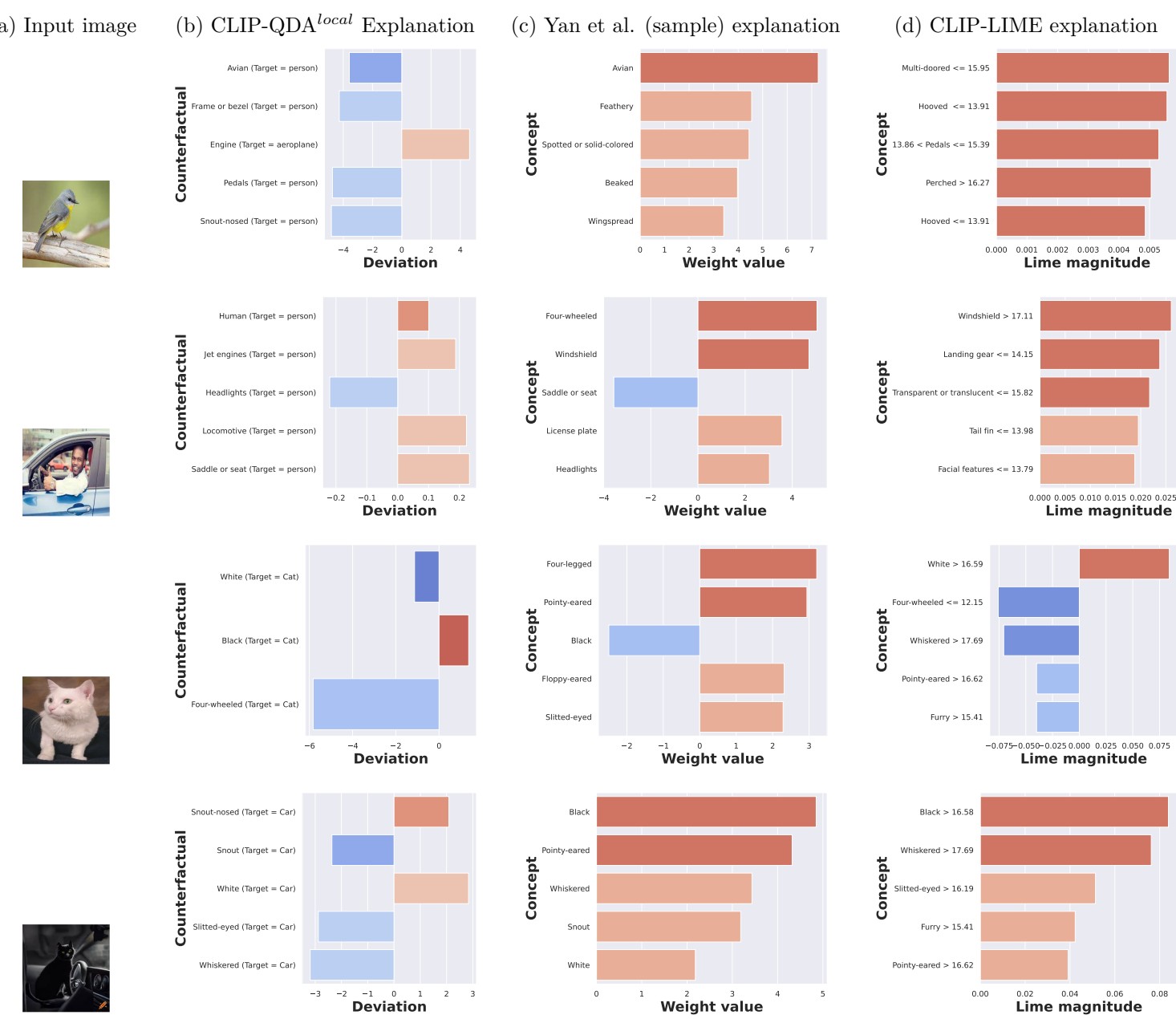

Figure 20: **Sample-wise explanations (concept level).** The first two examples come from the Pascal-PART dataset, and the last two samples come from the Cats/Dogs/Cars dataset in the biased setup. Note that the classifier mislabeled the $2^{nd}$ example as "car" and the $4^{th}$ example as "car".

(a) Input image                                   (b) CLIP-SHAP Explanation

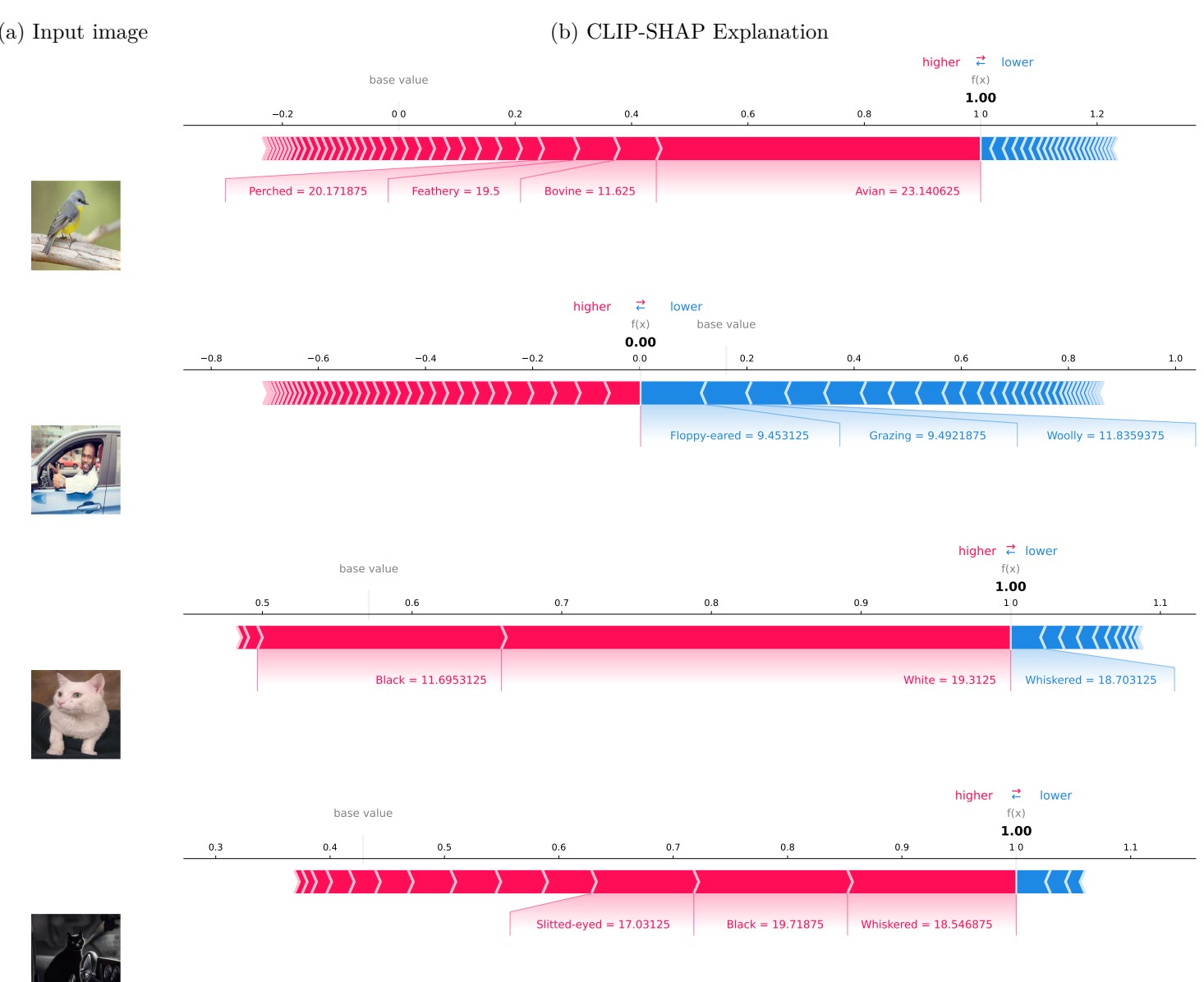

Figure 21: **Sample-wise explanations (concept level).** The first two examples come from the Pascal-PART dataset, and the last two samples come from the Cats/Dogs/Cars dataset in the biased setup. Note that the classifier mislabeled the $2^{nd}$ example as "car" and the $4^{th}$ example as "car".

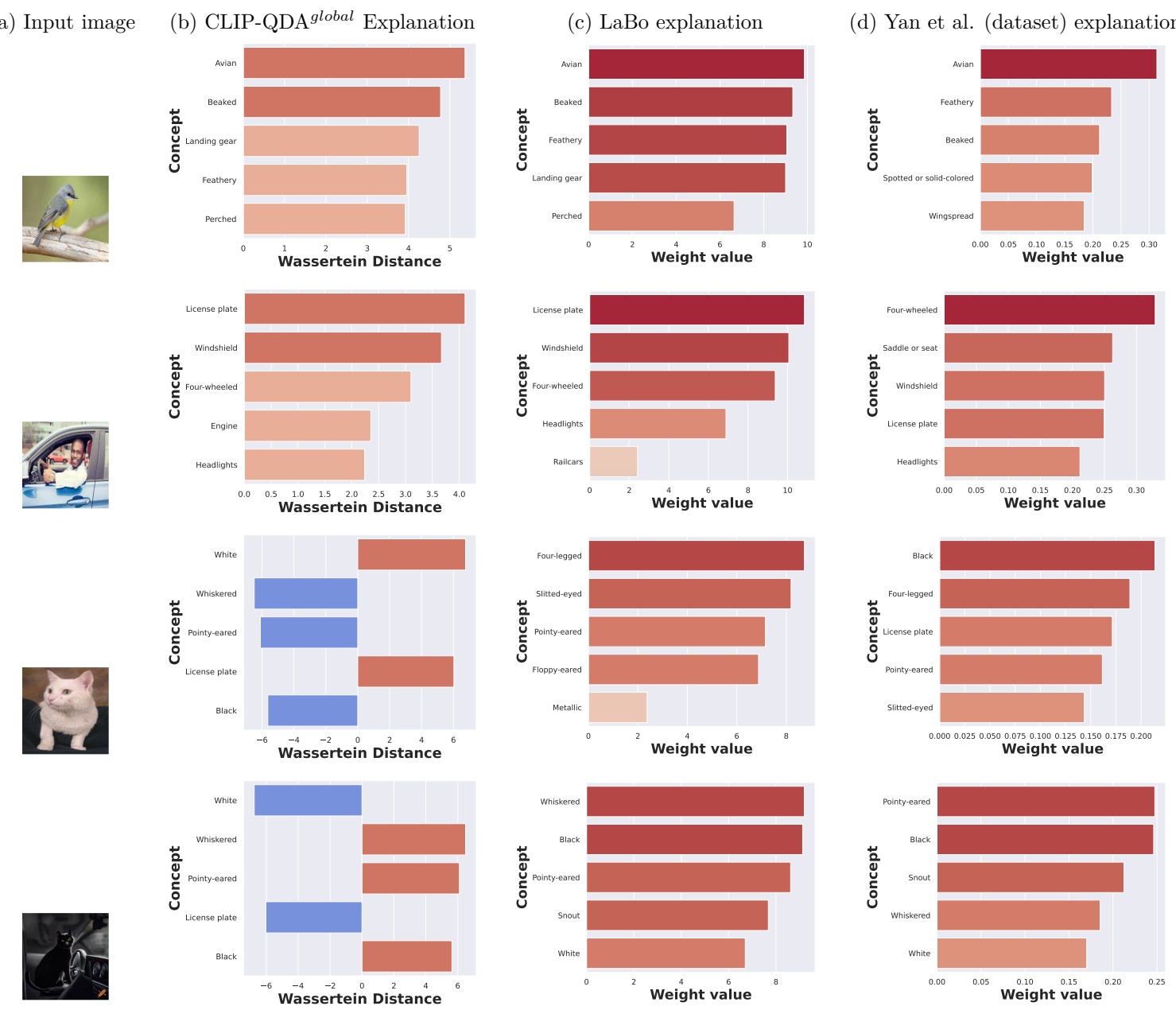

Figure 22: **Dataset-wise explanations.** The first two examples come from the PascalPART dataset, and the last two samples come from the Cats/Dogs/Cars dataset in the biased setup. Note that the classifier mislabeled the $2^{nd}$ example as "car" and the $4^{th}$ example as "car".

