# OpenReview forum: "CLIP-QDA: An Explainable Concept Bottleneck Model"
_TMLR — Accepted by TMLR_

### Review · Reviewer_SRtB · 2023-12-15

**Summary Of Contributions:**

### **Algorithm:**
This paper proposes a method for constructing explainable image classifiers from pre-trained CLIP backbones. The proposed method is based on the Concept Bottleneck Model (CBM), where the model's predictions are bottlenecked through a set of pre-defined concepts. This work in particular generates concept annotations for each image class by prompting an LLM. Next these concept words are encoded with a CLIP text encoder. Now to generate a class prediction for an image, the output of the CLIP image encoder is used to compute a cosine similarity score with each of the CLIP concept text embeddings. These CLIP scores are then fed to a QDA classifier, the parameters of which are estimated directly by computing statistics on the training data.

### **Assumptions:**
The QDA classifier makes the assumption that the conditional distribution of concept scores in the CLIP latent space (conditioned on the class label) are normally distributed, and therefore the CLIP latent space can be factored as a Mixture of Gaussians (MoG).

### **XAI Metrics:**
This work then proposes a Global XAI metric for explaining the classifier's predictions based on the Wasserstein-2 distance, which provides an indication into how the distribution of a concept score changes when conditioned on one class (c_1) versus another (c_2). This is used to provide global insights into how discriminative a certain concept word is for discriminating between class c_1 and c_2. Based on the proposed MoG decomposition, this metric yields a closed-form solution.

This work also proposes a signed Local XAI metric for explaining how the classifier's prediction on a specific instance would have changed if its CLIP score with a certain concept word was manually perturbed.

**Audience:**

Yes

**Claims And Evidence:**

Yes

**Requested Changes:**

I would suggest the following revisions to improve the work:
* Broaden the discussion of related work to describe the outline the literature in more detail than what is currently provided (see above).
* Broaden Table 1 to include more baselines, including at least
  * LaBo, SHAP, LIME
  * and provide details in the appendix about how these baselines are computed/obtained.
* Include ImageNet classification results, which are claimed to be included in the experimental protocol in the introduction, but cannot be found in the paper.
* Provide missing details about experimental protocol (see above), and please include in the main paper the strategy used for generating concept words (LLM prompting).

**Strengths And Weaknesses:**

**Strengths:**
* This work appears to be technically sounds and would certainly be of interest to certain individuals in TMLR's audience.

**Weaknesses:**
* Contextualization of this work in the literature and comparisons with related methods.
* Details about experimental protocol.
* Limited numerical results.

See more details below.

Strengths:
* Relevance for TMLR audience: CLIP backbones are widely utilized in the machine learning community, and thus a simple method for improving the explainability of CLIP image classifiers, post-hoc, is of interest to the TMLR community in my opinion. Moreover, this work builds on a growing body of work in machine learning revisiting CBMs in the context of multi-modal foundation models, which is likely of independent interest to the TMLR community.
* Simplicity: The proposed CLIP-QDA model is conceptually simple compared to other model-agnostic approaches for by-design or post-hoc explainability, such as classical shapely value estimation methods, which require re-training several classifiers, each time holding out a different subset of features. Moreover, the proposed explainability metrics are conceptually simple and yield closed-form solutions.
* Limitations: This work clearly discusses limitations of their proposed method by discussing where assumptions break down and their impact on performance.

Weaknesses:
* The biggest weakness is the situation of this work in the broader literature, both in discussion and numerical experiments.
  * For instance, no mention is made of post-hoc explainability methods based on explanation generation, such as GradCAM and related approaches.
  * Notably missing is also a discussion of Additive Feature Attribution methods, which are closely related to the proposed CLIP-QDA in that they seek to define the importance of specific features in latent space for predictions made by light-weight classifier heads.
  * Very related to this work is the LaBo work of Yang et al., 2023, which proposes to use an LLM to generate concept words for each image class, and then trains a linear classifier on top of the CLIP concept-word similarity scores, rather than a QDA classifier as proposed in this work.
  * What about a comparison with CounTEX, which also leverages a CLIP text encoder along with predefined concepts to produce concept activation vectors instead of concept scores.
  * What about model-agnostic approaches, such as Post-hoc Concept Bottleneck Models (PCBMs), which rely on a multi-modal models to obtain concept vectors, and then project the visual encoder’s embeddings onto this space, and thus can also be applied to image backbones which are not text aligned.
* Details about experimental protocol:
  * The paper lists ImageNet classification results, but I could not find these in the main paper, nor could I find them in the appendix.
  * Missing major details; what is the architecture of the CLIP backbone used in the considered experiments?
* Numerical results are currently limited to Table 1, showing image classification accuracy on PASCAL-Part, MIT scenes, Monoumi with limited baselines, and Figure 11, which shows deletion metric plots compared to other XAI methods.
* Results:
  * In terms of classification accuracy: CLIP linear probing is better than CLIP-QDA as number of concepts increases, which seems to correlated with the observation that the Gaussian assumption starts to break down with a larger number of concept words. But best performance for all models is obtained with increasing number of concept classes.
  * In terms of explainability, numerical results based on the deletion metric (how much does nullifying a certain number of concepts decrease classifier accuracy; thereby indicating the important concepts for classification) show that the proposed metric is significantly worse than the current state-of-the-art, SHAP, although the proposed method is much lighter to run; e.g., 45 minutes instead of 120 minutes on the largest dataset considered (PASCAL-Part).

---

### Review · Reviewer_czSS · 2024-01-01

**Summary Of Contributions:**

This paper introduces a new interpretable classifier based on CLIP. In particular, the authors suggest to model latent distributions with a mixture of gaussians, allowing to classify with a classic ML method such as quadratic discriminant analysis. Moreover, the authors propose global and local metrics to assess the learned representations based on measuring the distance between classes, and finding the minimal code that changes classification. Finally, the proposed approach is evaluated on basic examples, compared to basic baselines and analyzed.

**Audience:**

Yes

**Claims And Evidence:**

Yes

**Requested Changes:**

See above

**Strengths And Weaknesses:**

I am not an expert on CLIP-based modeling and Concept Bottleneck Models.

I like that the approach is straightforward, does not require training, and includes interpretability features. The mixture of gaussians seems like a nice assumption, however, it probably does not generalize to real-world cases, as noted in the paper. The paper is very readable and easy to follow. The analysis in Fig. 5 & 7 was nice.

I believe the approach could be strengthened by learning CLIP distributions, and not assuming a particular distribution. Alternatively, you could also try to guide CLIP towards GMM, so that your assumption holds. The evaluation section is somewhat unconvincing. The test cases and baselines seem too basic. Moreover, CLIP-QDA does not improve even in these basic examples.

---

### Review · Reviewer_xNTY · 2024-03-11

**Summary Of Contributions:**

This paper presents an interpretability tool that uses CLIP for image classification. Specifically, this paper uses a Mixture of Gaussians to model the latent distribution where each distribution can be interpreted as a concept. Based on this concept, they interpret the model behavior in terms of local and global behavior. Furthermore, they extend well-founded interpretable methods, including LIME and SHAP, to multi-modal setup by combining them with the proposed method.

**Audience:**

Yes

**Broader Impact Concerns:**

No ethical concern found

**Claims And Evidence:**

Yes

**Requested Changes:**

Visualizing the examples that are the centroid of each Gaussain distribution or giving some visualization of images inside the cluster (i.e., each Gaussain distribution)

**Strengths And Weaknesses:**

**Strength**

Overall, the writing is clear, and the paper is well-structured.

The method is sound i) using mixture of Gaussian to model the latent and using QDA to measure the concept probability, ii) decomposing the explanation into local and global.

Well-tackled the limitations of prior CLIP-based concept-based methods (i.e., requires dedicated datasets).

------------

**Weakness**

I think it is somewhat hard to understand the definition of 'concept' in this setup. I believe the concept of interpretable machine learning indicates two things: (i) human interpretable things that are used (ii) to decompose the reasoning process (of the prediction).
For instance, one example image of a bird contains a "beak" and "feather."\
However, since this paper does not use dedicated datasets, the proposed concept is only used for decomposability. I think explicitly adding some clusters of concepts will be helpful and labeling them with human experts will be useful.

---

> ### Author Response · Authors · 2024-03-23
> **Answers to xNTY**
>
> Thank you for your thoughtful review. We have made two adjustments to address your concerns, outlined below:
>
> 1 Clarification of the notion of concept:
>
> In our study, we have chosen to use the concept set generated by LLM prompting, drawing inspiration from LaBo's methodology. However, it's important to note that this approach can be adapted to incorporate additional prompting techniques or human-created datasets, opening various industrial applications. Initially, we explored incorporating parts as concepts provided by experts for pascalPART and the architectural characteristics for MonuMAI. However, after careful evaluation, we determined that the current setup yielded superior results. Also, to enhance clarity, we have included an additional categorization of concepts for each dataset, organized into subcategories. This information has been added to the appendix for reference.
>
> 2 Visualization of the centroid:
>
> In response to your feedback, we have conducted additional experiments focusing on visualizing samples relative to the distance of the centroid. Specifically, we examined samples that maximize and minimize the Mahalanobis distance between the sample of interest and the Gaussian prior associated with each class. Comprehensive details and results of these experiments are now available in Appendix A.3 for further insight.

---

### Author Response · Authors · 2024-02-16
**General answer**

Dear Reviewer and Meta reviewer,

Thank you for serving as the Meta reviewer and reviewer for our submission.  Thank you and the reviewers for the detailed feedback. We've decided to respond to the two first reviewers now and will address the last one once his/her reviews are released.

In our revised manuscript, we've made significant changes:

- Clarified contributions by including CLIP-SHAP and CLIP-LIME.
- Improved the evaluation protocol.
- Extended the number of baselines.
- Added quantitative and qualitative results.
- Enhanced the background and related works sections.
- Added implementation details.

All changes in the revision are highlighted in purple.

Sincerely,

---

### Comment · Action_Editor_L3xX · 2024-03-22
**Response to last review**

Dear Authors,

As the discussion period is coming to a close, please submit your response to the third review.

Thank you,
AE

---

### Author Response · Authors · 2024-03-23
**General answer: Update**

Dear Reviewer and Meta reviewer,

Following your previous comments and the new reviews, we have revised our manuscript accordingly. Specifically, we have expanded on the rationale behind our choice of concepts and included supplementary experiments focusing on visualizing samples concerning modeling the latent space.

---

### Author Response · Authors · 2024-05-27
**Revised version**

Dear Reviewers and Editor,

We thank you for your insightful feedback on our work. In response to the revisions requested, we have made several changes outlined below:

- Performance on ImageNet: We have revised our comments on the underwhelming accuracy performance on ImageNet. Specifically, we included additional remarks in the experiments section to explain the observed drop in accuracy on this dataset.
Figures 15 to 19: We have improved the captions of these figures and corrected typographical errors.

- Baselines: We have added references to the used repositories that we did not implement ourselves. In addition, the code will be made publicly available after publication. Additionally, we have reformulated the relevant section in the appendix for clarity.

- Clarity and Organization: We have reorganized the "Experiments" section to better separate the experimental setup from the experiments themselves, reducing redundancies. The revised version now makes a clearer distinction between the evaluations of our different contributions. Aside from some corrected typographical errors, the content remains the same.

We hope these changes address your concerns and enhance the clarity and comprehensiveness of our paper.

---

### Decision · Action_Editor_L3xX · 2024-04-29

**Recommendation:** Accept with minor revision

**Comment:**

The reviewers found the method interesting and simple. They raised questions in terms of the performance of the nroposed approach, especially for the ImageNet example. A reviewer also mentioned that the clarity of the paper has decreased after revision.

I would hence recommend that the authors ensure that the text correspond to the results presented, especially when they mention the performance of the method. Proof-reading might help with clarity, as well as including links to code repositories used in the work, adding details about the baselines, and improving on the description of figures 15-19 in appendix.

**Audience:**

The reviewers noted that the strength of the method is its simplicity. Therefore, despite some limitations of the proposed approach in terms of accuracy, I believe this work is of interest to the community.

**Claims And Evidence:**

The method proposed is sound and the authors added baselines during the rebuttal period which has strengthened the work. The reviewers have reported that some of the results were underwhelming in terms of accuracy, but I trust the authors will be careful in the text and abstract to provide nuance.